# STRAP: Spatio-Temporal Pattern Retrieval for Out-of-Distribution Generalization

**Haoyu Zhang**[▲♠▼*], **Wentao Zhang**[♣*], **Hao Miao**[♦*], **Xinke Jiang**[★], **Yuchen Fang**[★], **Yifan Zhang**[♠†]

▲ City University of Hong Kong, Hong Kong, China
♠ City University of Hong Kong (Dongguan), Guangdong, China
♣ Northeastern University, Shenyang, China
♦ The Hong Kong Polytechnic University, Hong Kong, China
★ University of Electronic Science and Technology of China, Chengdu, China
▼ SLAI, Shenzhen, China

hzhang2838-c@my.cityu.edu.hk    wentaozh2001@gmail.com    hao.miao@polyu.edu.hk
thinkerjiang@foxmail.com    fyclmiss@gmail.com    yifan.zhang@cityu-dg.edu.cn

## Abstract

Spatio-Temporal Graph Neural Networks (STGNNs) have emerged as a powerful tool for modeling dynamic graph-structured data across diverse domains. However, they often fail to generalize in Spatio-Temporal Out-of-Distribution (STOOD) scenarios, where both temporal dynamics and spatial structures evolve beyond the training distribution. To address this problem, we propose an innovative **S**patio-**T**emporal **R**etrieval-**A**ugmented **P**attern Learning framework, **STRAP**, which enhances model generalization by integrating retrieval-augmented learning into the STGNN continue learning pipeline. The core of STRAP is a compact and expressive pattern library that stores representative spatio-temporal patterns enriched with historical, structural, and semantic information, which is obtained and optimized during the training phase. During inference, STRAP retrieves relevant patterns from this library based on similarity to the current input and injects them into the model via a plug-and-play prompting mechanism. This not only strengthens spatio-temporal representations but also mitigates catastrophic forgetting. Moreover, STRAP introduces a knowledge-balancing objective to harmonize new information with retrieved knowledge. Extensive experiments across multiple real-world streaming graph datasets show that STRAP consistently outperforms state-of-the-art STGNN baselines on STOOD tasks, demonstrating its robustness, adaptability, and strong generalization capability without task-specific fine-tuning.

## 1 Introduction

**Spatio-Temporal Graph Neural Networks (STGNNs)** [36, 78, 43, 53, 35] have emerged as a powerful paradigm for modeling complex systems that evolve over both space and time. By integrating spatial and temporal dependencies [7, 29, 12] and modeling techniques [34, 46], STGNNs have surpassed traditional spatio-temporal approaches [36, 55], making extraordinary progress in diverse domains [72], such as traffic forecasting [66, 41], climate and weather prediction [53, 43], financial modeling [22, 28], and public health [31, 67]. However, training and deploying STGNNs on real-world **streaming** spatio-temporal data poses critical generalization challenges due to evolving dynamics across multiple dimensions: ❶ **Spatially**, systems undergo restructuring through node additions and removals [51, 55], leading to dynamic topologies and heterogeneity [76, 74, 14, 73]. ❷ **Temporally**, data exhibits periodic

---

*Indicates equal contribution.
†Corresponding author.

fluctuations [30], abrupt changes [5], and long-term drifts [62]. ❸ **Spatio-temporally**, coupling between space and time also produces synergistic impact and further results in joint distribution shifts [75, 59, 25], which may significantly impair STGNN generalization in dynamic scenarios.

This highly dynamic nature of spatio-temporal data causes traditional STGNN models to face severe **STOOD** (**S**patio-**T**emporal **O**ut-**O**f-**D**istribution) problems, leading to degraded performance for conventional models [82]. Existing efforts solve this critical problem as follows [68]: ❶ *Backbone-based methods* (Pretrain, Retrain), which either directly apply a model trained on historical data to new data without further training (Pretrain) or completely retrain the whole model from scratch using the new data (Retrain); ❷ *Architecture-based methods* (TrafficStream [10], ST-LoRA [54], STKEC [63], EAC [9]), which modify model architectures; ❸ *Regularization-based methods* like EWC [42], which constrain model parameter updates; ❹ *Replay-based methods* (Replay [16]), which reuse historical samples. Despite their progress, these approaches face the following limitations: backbone methods suffer catastrophic forgetting [40, 60], architecture-based approaches struggle with stability-plasticity trade-offs [58], regularization methods over-constrain adaptation [23], and replay techniques fail to distinguish between relevant and irrelevant historical knowledge [81, 32]. The root cause of the above deficiency is the insufficient exploitation of historical information and current information. Considering the distribution shifts in spatio-temporal graph data, historical information only partially benefits current predictions [70, 15], while some may introduce noise and even negative impacts. **Thus, the key challenge remains identifying which historical knowledge components provide the most valuable information gain for current predictions under complex spatio-temporal distribution shifts [8, 21].**

Instead of learning from the historical data and overwriting the model parameters to store the patterns implicitly, we propose to explicitly store the key similar patterns found in historical data to overcome the above limitations. The explicit storage can largely help to keep more historical information without memorizing it by updating the model parameters. Such external storage mechanisms are also capable of being cooperated with any STGNN-based backbones to enhance their ability to address STOOD problems. To this end, we need to handle the following challenges:

**C1. How to Effectively Identify and Efficiently Store Contributive Patterns?** Current approaches either concentrate solely on spatial aspects or rely on fixed spatio-temporal graphs trained on static datasets through updating the model parameters. For one thing, such a narrow focus prevents them from capturing richer and more complete historical patterns that could provide significant information gain. In addition, the information memorized in the model parameters is limited, especially for the STOOD tasks. Effectively identifying the most informative spatio-temporal patterns from historical data that maximize information gain for current predictions and storing these patterns efficiently are the cornerstones to solving the STOOD problem.

**C2. How to Optimally Balance the Integration of Historical Extracted Patterns with Current Observations?** Existing approaches face difficulties in accurately matching relevant patterns with the current observations due to insufficient similarity metrics and retrieval criteria. This limits the model's ability to fully exploit informative historical knowledge. On the other hand, excessive reliance on pattern matching may lead to overfitting to historical cases, blurring the boundary between prediction and retrieval. Therefore, a key challenge lies in developing mechanisms that can appropriately balance the incorporation of historical patterns with current data, ensuring both flexibility and robustness in evolving spatio-temporal prediction tasks.

To address the two aforementioned challenges, we propose the **S**patio-**T**emporal **R**etrieval-**A**ugmented **P**attern Learning (**STRAP**) framework. We construct a pattern library with a three-dimensional key-value architecture, where pattern keys serve as efficient retrieval indices and pattern values encapsulate rich contextual information.

At its core, STRAP maintains specialized collections for spatial, temporal, and spatio-temporal patterns. Pattern keys are extracted using geometric topological and properties (e.g., graph curvature, clustering coefficients) and time series characteristics (e.g., wavelet transformations), optimized for similarity matching. Pattern values, generated through a dedicated STGNN backbone, preserve the essential features needed for downstream tasks. Specifically, STRAP employs a two-phase mechanism: (1) similarity-based retrieval that matches current graph features with historical pattern keys, and (2) feature enhancement through fusion of retrieved pattern values with current representations. As such, *C1* is addressed by constructing and updating the external pattern library based on past retrieval values, and retrieving the similar patterns from the library for the downstream tasks. Furthermore, to solve *C2*, our approach for spatio-temporal pattern extraction captures cross-dimensional dependencies,

while an adaptive fusion and training mechanism calibrates the influence of historical patterns. In summary, our key contributions are as follows:

- We propose STRAP, a novel plug and play framework tailored for STOOD scenarios, which constructs a key-value pattern library that captures multi-dimensional patterns across spatial, temporal, and spatio-temporal domains, fundamentally decoupling pattern indexing from pattern utilization to mitigate catastrophic forgetting and alleviate the STOOD problem.
- We design an adaptive fusion and learning mechanism that dynamically integrates retrieved historical patterns with current observations, enabling robust and flexible prediction over continuously evolving spatio-temporal data streams.
- Experiments on multiple real-world streaming graph datasets demonstrate that STRAP achieves SOTA performance on STOOD tasks, showcasing its effectiveness in continual generalization.

## 2 Related Work

### 2.1 Continue Learning on STOOD Tasks

Continuous Learning typically maintains long-term and important information while updating model memory using newly arrived instances [52, 4, 9]. Due to continuous learning's excellent performance in adapting to evolving data and avoiding catastrophic forgetting, a series of methods based on continuous learning have been proposed. Chen [10] introduced a historical data replay strategy called TrafficStream based on the classic replay strategy in continuous learning, which feeds all nodes to update the neural network, thereby achieving a balance between historical information and current information. Similarly based on experience replay, Wang et al. [64] proposed PECPM, a continuous spatio-temporal learning framework based on pattern matching, which learns to match patterns on evolving traffic networks for current data traffic prediction. Miao et al. [48] further emphasized replay-based continuous learning by sampling current spatio-temporal data and fusing it with selected samples from a replay buffer of previously learned observations. Chen et al. [10] proposed a parameter-tuning prediction framework called EAC, which freezes the base STGNN model to prevent knowledge forgetting and adjusts prompt parameter pools to adapt to emerging expanded node data.

### 2.2 Pattern Retrieval Learning

Pattern Retrieval Learning refers to methods of representation learning and reasoning through identifying, extracting, and utilizing key patterns in data [44, 57, 65, 19]. In the context of graph neural networks, pattern retrieval learning typically focuses on how to discover stable and predictive patterns from graph structures and node features [3, 27, 17]. In spatio-temporal data, pattern retrieval faces unique challenges as it must simultaneously consider spatial dependencies and temporal evolution patterns [2]. Han et al. [20] proposed RAFT, a retrieval-augmented time series forecasting method that directly retrieves historical patterns most similar to the input from training data and utilizes their future values alongside the input for prediction, reducing the model's burden of memorizing all complex patterns. Li et al. [36] proposed a framework focused on extracting seasonal and trend patterns from spatio-temporal data. By representing these patterns in a disentangled manner, their method could better handle distribution shifts in non-stationary temporal data. In the area of out-of-distribution detection, Zhang et al. [79] proposed an OOD detection method based on Modern Hopfield Energy, which memorizes in-distribution data patterns from the training set and then compares unseen samples with stored patterns to detect out-of-distribution samples,but it's heavy reliance on attention mechanisms limits it's stability in complex and highly variable environments.

## 3 Preliminaries

We first formalize spatio-temporal graph [1, 78] data structures, followed by an introduction to OOD learning with a focus on spatial and temporal distribution shifts. Based on these foundations, we propose a unified prediction framework designed to address the challenges of robust modeling in dynamically evolving environments. Notations are provided in in Table 3 in Appendix A.

***Definition 1. (Dynamic Spatio-Temporal Graph)*** We define a dynamic spatio-temporal graph as a time-indexed sequence of graphs $\mathcal{G}_t = (\mathcal{V}_t, \mathbf{A}_t), t \in [1, T]$, where $\mathcal{V}_t = \{v_1, v_2, ..., v_N\}$ is the set of

$N_t$ nodes and $\mathbf{A}_t \in \mathbb{R}^{N \times N}$ is the weighted adjacency matrix at time step $t$, encoding dynamic pairwise relationships among nodes. Each entry $\mathbf{A}_{t,ij} \in \mathbb{R}$ indicates the strength of the connection between nodes $v_i$ and $v_j$ at time $t$, and its size $N_t$, which may vary over time. At each discrete time step $t$, a graph signal $\mathbf{X}_t \in \mathbb{R}^{N \times c}$ is observed, with each node is associated with a $c$-dimensional feature vector. The sequence of graph signals over $T$ time steps is denoted as $\mathbf{X} = [\mathbf{X}_1, \mathbf{X}_2, ..., \mathbf{X}_T] \in \mathbb{R}^{T \times N \times c}$.

***Definition 2. (Spatio-Temporal Out-of-Distribution Learning)*** Let the training environment be denoted as $e^{\mathrm{tr}} = (\mathcal{G}^{\mathrm{tr}}, \mathcal{D}^{\mathrm{tr}})$, where $\mathcal{G}^{\mathrm{tr}} = (\mathcal{V}^{\mathrm{tr}}, \mathbf{A}^{\mathrm{tr}})$ represents the training graph and $\mathcal{D}^{\mathrm{tr}} = \{(\mathbf{X}^{(i)}, \mathbf{Y}^{(i)})\}_{i=1}^{n}$ is the corresponding dataset consisting of $n$ samples. The objective of OOD learning [62, 79] is to generalize from this training environment to arbitrary test environments $e^{\mathrm{te}} \sim \mathcal{E}$, where test distributions may differ significantly from those seen during training, i.e., $P(e^{\mathrm{te}}) \neq P(e^{\mathrm{tr}})$. This formulation accommodates both classic OOD scenarios: where test environments differ significantly from training due to structural or temporal changes and continual learning scenarios: where distribution shifts emerge progressively as new data streams in over time.

In STRAP, we consider three types of distribution shifts: ❶ **Spatial distribution shifts:** Changes in the underlying graph structure, formally denoted as $P(\mathcal{G}^{\mathrm{te}}) \neq P(\mathcal{G}^{\mathrm{tr}})$. These shifts may arise from variations in node connectivity, edge weights, or even the addition or removal of nodes. Such changes affect how information flows through the network and can alter the relative importance of nodes and edges. ❷ **Temporal distribution shifts:** Changes in temporal dynamics, expressed as $P(\mathbf{X}_{t+1}|\mathbf{X}_{1:t})^{\mathrm{te}} \neq P(\mathbf{X}_{t+1}|\mathbf{X}_{1:t})^{\mathrm{tr}}$. These shifts reflect evolving temporal patterns, including modifications in periodicity, trend, or correlation structure. They can emerge due to abrupt regime changes or through gradual evolution over time. ❸ **Spatio-temporal distribution shifts:** Joint changes across both spatial and temporal dimensions, captured by $P(\mathbf{X}_{t+1}|\mathbf{X}_{1:t}, \mathcal{G}_{1:t+1})^{\mathrm{te}} \neq P(\mathbf{X}_{t+1}|\mathbf{X}_{1:t}, \mathcal{G}_{1:t+1})^{\mathrm{tr}}$. These shifts model the interplay between dynamic graph structures and evolving temporal signals, often arising in real-world settings where the data-generating process itself is non-stationary and context-dependent.

***Definition 3. (Unified STOOD Prediction Framework)*** STOOD prediction framework addresses the challenge of generalizing across both spatial and temporal distribution shifts by formulating the learning objective as a unified min-max optimization problem [9, 10]:

$$\Theta^* = \operatorname*{argmin}_{\Theta} \sup_{e \in \mathcal{E}} \mathbb{E}_{(\mathbf{X}, \mathbf{Y}) \sim P(e)}[\mathcal{L}(f_\Theta(\mathbf{X}), \mathbf{Y})], \tag{1}$$

Where $f_\Theta : \mathbb{R}^{T \times N \times c} \rightarrow \mathbb{R}^{T_p \times N \times c}$ is a neural network parameterized by $\Theta$, which maps an input sequence of node features to a sequence of future predictions. The target output $\mathbf{Y} = [\mathbf{X}_{T+1}, \mathbf{X}_{T+2}, ..., \mathbf{X}_{T+T_p}] \in \mathbb{R}^{T_p \times N \times c}$ represents the ground-truth observations over a prediction horizon of $T_p$ time steps. The loss function $\mathcal{L}$ measures the gap between predicted and true values, typically using MSE for regression or Binary Cross Entropy for classification tasks. The conditional distribution $P(e)$ characterizes the data generation in environment $e$, while $\mathcal{E}$ represents all possible environments, including those with distributions different from the training environment.

## 4 STRAP Framework

We present STRAP, a retrieval-augmented framework addressing streaming spatio-temporal outlier detection by leveraging historical patterns (Figure 1). Our approach first decomposes complex spatio-temporal data into manageable subgraphs, then extracts and stores multi-dimensional patterns in a searchable library. During inference, STRAP retrieves relevant historical patterns and adaptively fuses them with current observations for optimal prediction. For enhanced clarity, the Spatio-Temporal Pattern Library Construction is outlined in Algorithm 1 (cf. Appendix B.2) and the Training and Inference with Toy Graphs Retrieval are detailed in Algorithm 2 (cf. Appendix B.2). Theoretical analysis in Appendix B.1 demonstrates the effectiveness of our pattern extraction and retrieval mechanisms under distribution shift.

### 4.1 Spatial Temporal Subgraph Chunking

To effectively capture diverse and disentangled spatio-temporal patterns, we decompose the original graph sequence into three types of specialized subgraphs, each designed to emphasize a specific dimension of variability: spatial, temporal, or spatio-temporal.

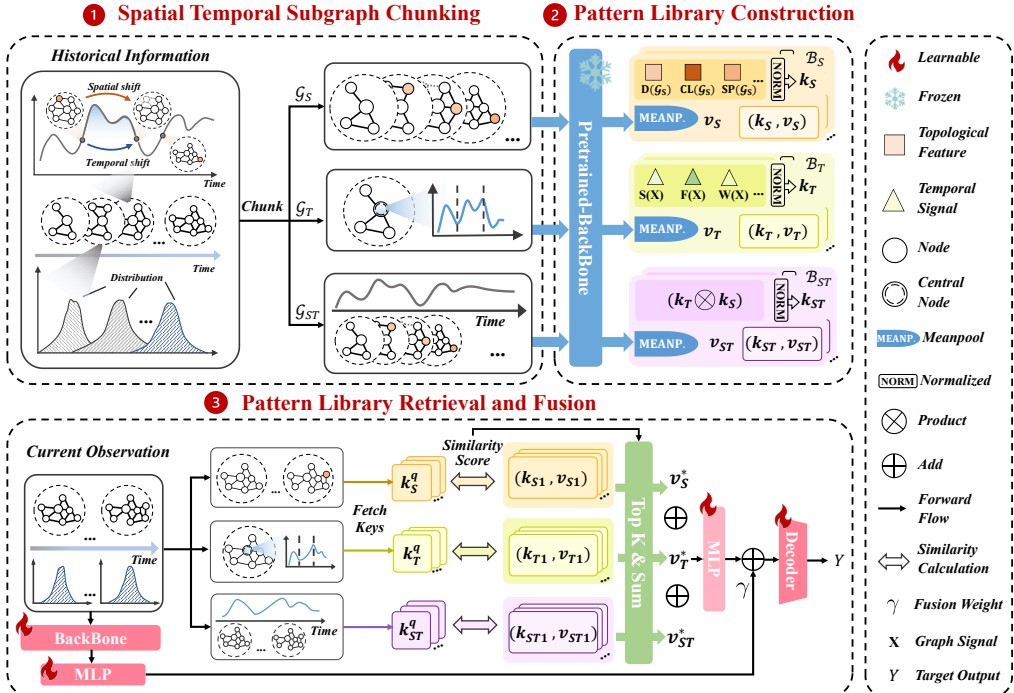

Figure 1: The overall framework of STRAP.

**❶ Spatial Subgraphs** ($\mathcal{G}_S$): These subgraphs $\{\mathcal{G}_S\}$ are constructed by slicing the temporal axis into overlapping windows of fixed time length $\tau_S$. For each window $[t, t + \tau_S]$, we aggregate both the connectivity and node features to form a window-level spatial representation:

$$\mathcal{G}_S = (\mathcal{V}, \bar{\mathbf{A}}_{[t,t+\tau_S]}, \bar{\mathbf{X}}_{[t,t+\tau_S]}) \tag{2}$$

Where $\bar{\mathbf{A}}_{[t,t+\tau_S]}$ is the time-averaged adjacency matrix, and $\bar{\mathbf{X}}_{[t,t+\tau_S]}$ is the aggregated node feature matrix across the window. These subgraphs emphasize persistent spatial structures that remain stable over short temporal intervals.

**❷ Temporal Subgraphs** ($\mathcal{G}_T$): To isolate temporal dynamics, we sample a central node $v_c$ along with its $k$-hop spatial neighborhood $\mathcal{N}_k(v_c)$ during $T$, and track their evolution over the full time span:

$$\mathcal{G}_T = (\{v_c, \mathcal{N}_k(v_c)\}, \mathbf{A}_{[1:T]}|_{\{v_c, \mathcal{N}_k(v_c)\}}, \mathbf{X}_{[1:T]}|_{\{v_c, \mathcal{N}_k(v_c)\}}) \tag{3}$$

Where $\mathbf{A}_{[1:T]}|_{\mathcal{N}_k(v_c)}$ and $\mathbf{X}_{[1:T]}|_{\mathcal{N}_k(v_c)}$ are the sequences of adjacency matrices and node features restricted to $v_c$ and its neighbors $\mathcal{N}_k(v_c)$. These subgraphs $\{\mathcal{G}_T\}$ focus on localized temporal.

**❸ Spatio-Temporal Subgraphs** ($\mathcal{G}_{ST}$): To jointly capture spatial and temporal interactions, we first apply spectral clustering [61] to partition the node set into $m$ communities $\{\mathcal{C}_1, \mathcal{C}_2, ..., \mathcal{C}_m\}$ based on structural coherence. For each community $\mathcal{C}_i$, we extract time-windowed subgraphs:

$$\mathcal{G}_{ST} = (\mathcal{C}_i, \mathbf{A}_{[t,t+\tau_{ST}]}|_{\mathcal{C}_i}, \mathbf{X}_{[t,t+\tau_{ST}]}|_{\mathcal{C}_i}) \tag{4}$$

Where the adjacency matrices and node features are restricted to $\mathcal{C}_i$ within the interval $[t, t + \tau_{ST}]$. These subgraphs $\{\mathcal{G}_{ST}\}$ capture coherent spatio-temporal dynamics among functionally related node groups.

## 4.2 Multi-dimensional Pattern Library Construction

To more effectively capture dynamic spatio-temporal patterns, we extend the chunking approach introduced in Section 4.1 by constructing the structured key-value pattern libraries across different dimensions. Each subgraph instance—spatial, temporal, or spatio-temporal—is encoded as a key-value pair that captures characteristic variations within its respective dimension. The processes for generating pattern keys and values are detailed in Sections 4.2.1 and 4.2.2, respectively. Based on these patterns, we construct three distinct pattern libraries: **❶ Spatial Library** $\mathcal{B}_S$, **❷ Temporal Library** $\mathcal{B}_T$, and **❸ Spatio-Temporal Library** $\mathcal{B}_{ST}$.

### 4.2.1 Pattern Key Generation

Retrieval-augmented learning effectiveness depends on discriminative pattern keys robust to distributional shifts, for which we design dimension-specific extraction methods.

**Spatial Pattern Keys.** Spatial shifts often manifest as changes in graph connectivity, community composition, or centrality structure [13]. To capture them, we extract multi-scale topological features:

❶ *From the aspect of local structure*, we compute neighborhood statistics $\mathbf{D}(\mathcal{G}_S) = [\mu_d, \sigma_d, \max(d), \min(d)]$ [6], where $d$ represents node degrees, and clustering coefficients [56] $\mathbf{CL}(\mathcal{G}_S) = [C_1, C_2, ..., C_n]$ with $C_i = \frac{2|\{e_{jk}\}|}{k_i(k_i-1)}$, $e_{jk}$ are edges between neighbors of node $i$, and $k_i$ is the degree of node $i$. ❷ *From the aspect of global connectivity,* For robustness against changing graph scales, we calculate path-based metrics such as shortest path statistics [45] $\mathbf{SP}(\mathcal{G}_S) = [\mu_{\text{sp}}, \sigma_{\text{sp}}, \text{diam}(\mathcal{G}_S)]$ where $\mu_{\text{sp}}$ and $\sigma_{\text{sp}}$ are the mean and standard deviation of shortest path lengths, and $\text{diam}(\mathcal{G}_S)$ is the graph diameter. ❸ *From the aspect of geometric properties,* We also employ Forman-Ricci curvature [33] to capture intrinsic geometric properties that remain stable despite local perturbations. For a node $v_i$, this is computed as: $\mathbf{FR}(v_i) = 1 - \sum_{u_j \in \mathcal{N}(v_i)} \frac{d_{v_i} \cdot d_{u_j}}{2w_{e_{ij}}} + \sum_{e_{ij} \in E(v_i)} \frac{d_{v_i}}{w_{e_{ij}}}$. where $d_{v_i}$ is the degree of node $v_i$, $\mathcal{N}(v_i)$ is the set of neighboring nodes, $w_{e_{ij}}$ is the edge weight between nodes $v_i$ and $u_j$, and $E(v_i)$ is the set of incident edges. This curvature analysis identifies distinct topological structures: negative curvature nodes, zero curvature nodes, and positive curvature nodes.

Finally, we concatenate and normalize these features to form the spatial key:

$$\mathbf{k}_S = \text{NORMALIZE}([\mathbf{D}(\mathcal{G}_S); \mathbf{CL}(\mathcal{G}_S); \mathbf{SP}(\mathcal{G}_S); \mathbf{FR}(v)]) \tag{5}$$

**Temporal Pattern Keys.** Temporal shifts may arise from changes in variance, periodicity, trend, or complexity [47, 39, 38, 11]. To address this, we extract features that describe the temporal signal at multiple scales:

❶ *Statistical and spectral descriptors:* We compute statistical moments $\mathbf{S}(X) = [\mu_X, \sigma_X, \text{skew}(X), \text{kurt}(X)]$, and extract dominant frequency components $\mathbf{F}(X) = [\omega_1, ..., \omega_m, E_1, ..., E_m]$, where each $\omega_i$ denotes a prominent frequency and $E_i$ its corresponding energy. ❷ *Multi-resolution analysis:* To capture transient and multi-scale phenomena, we apply wavelet transforms [77]: $\mathbf{W}(X) = \left\{ W_\psi X(a,b) = \frac{1}{\sqrt{a}} \int_{-\infty}^{\infty} X(t) \psi^* \left( \frac{t-b}{a} \right) dt \right\}$ where $a$ is the scale parameter and $b$ the time shift. ❸ *Temporal dependencies and complexity:* We compute autocorrelation functions $\mathbf{R}(X) = \{R_{xx}(k)\}$ to identify periodicity at lag $k$, and entropy $\mathbf{H}(X) = \{-\sum_i p(x_i) \log p(x_i)\}$ to quantify uncertainty and regularity.

Finally, we concatenate and normalize these features to form the temporal key:

$$\mathbf{k}_T = \text{NORMALIZE}([\mathbf{S}(X); \mathbf{F}(X); \mathbf{W}(X); \mathbf{R}(X); \mathbf{H}(X)]) \tag{6}$$

**Spatio-Temporal Interaction Keys.** To encode joint patterns that emerge from the interaction of structure and dynamics, we define cross-dimensional interaction keys as:

$$\mathbf{k}_{ST} = \text{NORMALIZE}(\mathbf{k}_S \otimes \mathbf{k}_T) \tag{7}$$

Where $\otimes$ denotes an element-wise cross-product. This formulation allows the model to capture coupled dependencies across spatial and temporal dimensions without treating them independently.

### 4.2.2 Pattern Value Generation

Pattern values serve as the semantic content associated with each key, representing informative subgraph embeddings extracted from the backbone model. For $\mathcal{G}_i$ in any dimension $\mathcal{B}_{\cdot} \in \{\mathcal{B}_S, \mathcal{B}_T, \mathcal{B}_{ST}\}$, we compute the pattern value by applying a frozen shared pretrained STGNN backbone parameterized by $\Theta_{pt}$ (i.e., STGCN [69], ASTGCN [18]) to the corresponding subgraph and aggregating its pattern as $\mathbf{v}_i$:

$$\mathbf{v}_i = \text{MEANPOOL}(\text{BACKBONE}_{\Theta_{pt}}(\mathcal{G}_i)), \tag{8}$$

Where $\text{MEANPOOL}(\cdot)$ denotes mean-pooling to obtain the full $\mathcal{G}_i$ value. Thus, for each dimension, we store each key-value pair $(\mathbf{k}_i, \mathbf{v}_i)$ into the pattern library construction.

### 4.3 Pattern Retrieval and Knowledge Fusion

#### 4.3.1 Pattern Library Retrieval

After constructing and indexing the pattern libraries, we implement a similarity-based retrieval process to identify relevant historical patterns for the current observation. For $\mathcal{G}_i$ in any dimension $\mathcal{B}. \in \{\mathcal{B}_S, \mathcal{B}_T, \mathcal{B}_{ST}\}$, we first compute the similarity between the query key $\mathbf{k}_i^q$ (extracted from the current subgraph) and all stored keys $\mathbf{k}_j$ in the corresponding library $\mathcal{B}.$:

$$s(\mathbf{k}_i^q, \mathbf{k}_j) = \exp(-\|\mathbf{k}_i^q - \mathbf{k}_j\|_2), \quad \mathbf{k}_j \in \mathcal{B}. \tag{9}$$

Similarity scores are then used to retrieve the top-$k$ most relevant key-value pairs from each library:

$$\mathcal{R}_i = \left\{ (\mathbf{k}_j^i, \mathbf{v}_j^i, s_j^i) \,\big|\, j \in \mathrm{TopK}_{\mathcal{B}.} \ s(\mathbf{k}_i^q, \mathbf{k}_j) \right\} \tag{10}$$

Where $s_j^i$ denotes the similarity score, $\mathbf{k}_j^i$ is the retrieved pattern key, and $\mathbf{v}_j^i$ is its corresponding value.

By performing retrieval independently across spatial, temporal, and spatio-temporal libraries, the model obtains multiple sets of complementary pattern-value pairs. This multi-dimensional retrieval strategy enables the model to incorporate a diverse set of historical contexts—structural patterns from graph topology, temporal dynamics from time series behavior, and integrated patterns from their interaction—thereby improving its ability to make robust predictions under distributional shifts, including both abrupt changes and gradual evolutions.

#### 4.3.2 Knowledge Fusion Mechanism

After retrieving relevant pattern values from each library, we integrate this historical knowledge with the current observation using an information-theoretic fusion mechanism designed to maximize the joint representational capacity. For $\mathcal{G}_i$ in any dimension $\mathcal{B}. \in \{\mathcal{B}_S, \mathcal{B}_T, \mathcal{B}_{ST}\}$, we first compute a similarity-weighted average over the retrieved values: $\mathbf{v}_i^* = \sum_{(\mathbf{k}_j^i, \mathbf{v}_j^i, s_j^i) \in \mathcal{R}_i} \mathrm{SOFTMAX}(s_j^i) \cdot \mathbf{v}_j^i$.

Next, to capture non-linear cross-dimensional dependencies, we utilize the same architectural backbone, parameterized by $\Theta_{\mathrm{train}}$ (initialized from $\Theta_{\mathrm{pt}}$ and subsequently fine-tuned), to encode the current observation $\mathcal{G}$. Two separate multilayer perceptrons—$\mathrm{MLP}_1$ for the current observation and $\mathrm{MLP}_2$ for the retrieved values—are then applied to embed the respective representations. Formally, the transformations are defined as:

$$\mathbf{Z}_1 = \mathrm{MLP}_1\big(\mathrm{BACKBONE}_{\Theta_{\mathrm{train}}}(\mathcal{G}_i)\big), \quad \mathbf{Z}_2 = \mathrm{MLP}_2(\mathbf{v}_S^* \oplus \mathbf{v}_T^* \oplus \mathbf{v}_{ST}^*). \tag{11}$$

Next, the final fused representation is computed as a convex combination of the transformed current input and the aggregated historical knowledge:

$$\mathbf{Z} = \gamma \cdot \mathbf{Z}_1 + (1 - \gamma) \cdot \mathbf{Z}_2, \tag{12}$$

Where $\gamma$ is a fusion weight that calibrates the balance between current observations and retrieved historical patterns. The resulting fused embedding $\mathbf{Z}$ is then channeled into the decoder for downstream tasks. This deliberate integration mechanism synthesizes complementary information sources, allowing the model to leverage both immediate context and relevant historical knowledge, thereby enhancing predictive performance across varying distribution conditions.

## 5 Experiments

In this section, we present a comprehensive set of experiments to evaluate the effectiveness of STRAP compared to state-of-the-art baselines on three real-world streaming spatio-temporal graph datasets. Our experiments are designed to answer the following research questions:

- **RQ1:** How does STRAP perform compared to state-of-the-art methods on STOOD task?

- **RQ2:** What are the contribution of different pattern library and key components to the overall performance?

- **RQ3:** How sensitive is STRAP to key hyperparameters? (Results are presented in Appendix C.4.)

- **RQ4:** How robust is STRAP when deployed under significant distribution shifts, including both abrupt and gradual changes?
- **RQ5:** How does STRAP perform compared to other retrieval-based methods, and how does it perform in terms of computational efficiency (Results are presented in Appendix C.6.)?
- **RQ6:** How does STRAP perform compared to baseline methods in few-shot scenarios?

## 5.1 Experimental Setup

**Datasets.**   We evaluate STRAP on three real-world streaming spatio-temporal graph datasets: *AIR-Stream* [9], *PEMS-Stream* [10], and *ENERGY-Stream* [9]. Detailed dataset statistics, experimental settings and evaluation are provided in Table 8 in Appendix C.3 and C.1 in Appendix C.

**Backbones and Baselines.**   We consider four versions of our proposed framework STRAP: ❶ STRAP, which utilizes the complete pattern library system; ❷ STRAP w/o S, which indicates we operate without the spatial pattern library; ❸ STRAP w/o T, which operates without the temporal pattern library; and 4) STRAP w/o S+T, which functions without the spatio-temporal pattern library. For baselines, we implement: ❶ two standard training paradigms are included: Pretrain and Retrain, which directly utilize the backbone models; ❷ we also compare with state-of-the-art approaches for graph learning: TrafficStream [10], ST-LoRA [54], STKEC [63], EAC [9], EWC [42], Replay [16], ST-Adapter [49], GraphPro [71], and PECPM [64]. These baselines represent diverse strategies for handling graph-structured streaming data, including backbone, architecture-based, regularization-based, and replay-based methods for evolving graph streams. For STRAP variants and baselines, we conduct experiments with four different backbone architectures: STGNN [69], ASTGNN [18], DCRNN [37], and TGCN [80]. A detailed description of baselines and backbones can be referred to in Appendix C.2.

## 5.2 STRAP Results (RQ1 & RQ2)

As illustrated in Table 1, our STRAP framework consistently outperforms baseline methods across different datasets and metrics. We summarize our key findings below. Further details and experiment results are provided in Table 4, 5 and 6 in Appendix C.

**SOTA Result Across Categories (RQ1).**   **STRAP achieves the highest average performance with 7.17% improvement across metrics (MAE: 4.88%, RMSE: 9.12%, MAPE: 7.5%).** This stems from our multi-level pattern library approach that effectively captures complex streaming spatio-temporal dynamics. Our analysis reveals key advantages over existing categories: backbone-based methods lack OOD handling mechanisms, leading to catastrophic forgetting [70]; architecture-based and regularization-based approaches implement parameter protection but insufficiently utilize historical data; and replay-based methods store knowledge implicitly in model parameters, facing capacity constraints [50]. In contrast, STRAP addresses **C1** through explicit pattern libraries and retrieval-based fusion, maintaining an interpretable external memory that ensures robust performance across varying distribution conditions.

**Ablation Study (RQ2).**   The ablation studies (w/o S, w/o T, w/o S+T) clearly demonstrate each pattern library's importance, with the spatial component being most critical. The temporal component also significantly contributes to performance.

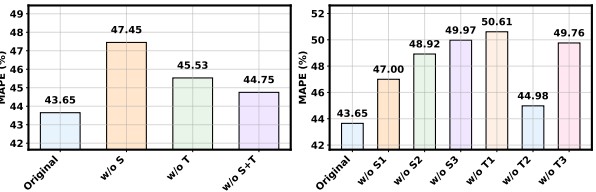

Figure 2: Impact of different pattern libraries and keys. Left: library. Right: key.

Specifically, Spatial 1 and 2 refer to features from the aspect of local structure and the shortest path statistics feature from global connectivity, while Spatial 3 represents the Forman-Ricci curvature from geometric properties. In terms of time, Temporal 1, 2, and 3 correspond to Statistical and spectral descriptors, Multi-resolution analysis, and Temporal dependencies and complexity, respectively. From the experimental results, in the spatial pattern analysis, **Spatial 3 (w/o S3) demonstrates the highest importance with a MAPE of 49.97%, representing a performance drop of 6.32% when removed**. This indicates that the Forman-Ricci curvature from geometric properties plays a crucial role in capturing

the intrinsic geometric structure of spatio-temporal graphs. Spatial 1 and 2, while contributing to the model performance, show relatively smaller impacts when removed individually, suggesting that geometric features may be more critical than previously assumed in spatio-temporal forecasting tasks. In the temporal pattern analysis, **Temporal 1 (w/o T1) emerges as the most critical component with a MAPE of 50.61%, showing the largest performance degradation of 7.96% when removed**. This demonstrates that Statistical and spectral descriptors are fundamental for temporal modeling. In contrast, Temporal 2 (Multi-resolution analysis) shows the smallest impact when removed (MAPE: 44.98%), indicating that while useful, it may be the least critical among the temporal components for this specific forecasting task.

Table 1: Comparison of the overall performance of different methods (STGNN backbone).

| | Datasets | | Air-Stream | | | | PEMS-Stream | | | | Energy-Stream | | | |
|---|---|---|---|---|---|---|---|---|---|---|---|---|---|---|
| Category | Method | Metric | 3 | 6 | 12 | Avg. | 3 | 6 | 12 | Avg. | 3 | 6 | 12 | Avg. |
| **Back-bone** | *Pretrain* | MAE | 18.96±2.55 | 21.87±2.15 | 25.02±1.59 | 21.62±2.15 | 14.06±0.18 | 15.14±0.19 | 17.44±0.24 | 15.32±0.20 | 10.71±0.05 | 10.74±0.09 | 10.76±0.10 | 10.73±0.08 |
| | | RMSE | 30.11±3.81 | 35.21±3.31 | 40.26±2.62 | 34.58±3.33 | 21.86±0.23 | 23.97±0.27 | 28.10±0.36 | 24.24±0.27 | 10.86±0.06 | 10.98±0.15 | 11.06±0.15 | 10.95±0.11 |
| | | MAPE (%) | 22.88±2.18 | 27.04±1.59 | 32.01±0.95 | 26.86±1.63 | 29.03±2.96 | 30.01±2.80 | 32.28±2.48 | 30.14±2.65 | 175.12±5.41 | 177.49±8.28 | 178.50±8.52 | 176.83±7.31 |
| | *Retrain* | MAE | 19.16±1.42 | 21.90±1.21 | 25.02±0.97 | 21.73±1.23 | 12.93±0.08 | 14.04±0.05 | 16.35±0.05 | 14.22±0.05 | 5.50±0.05 | 5.42±0.17 | 5.42±0.17 | 5.45±0.12 |
| | | RMSE | 30.13±1.95 | 34.88±1.60 | 39.89±1.30 | 34.42±1.67 | 20.86±0.09 | 22.94±0.06 | 26.98±0.11 | 23.19±0.08 | 5.66±0.05 | 5.64±0.15 | 5.74±0.15 | 5.67±0.09 |
| | | MAPE (%) | 24.98±2.74 | 28.69±2.32 | 33.16±1.71 | 28.53±2.27 | 18.75±0.51 | 20.12±0.39 | 23.39±0.39 | 20.44±0.42 | 52.22±0.18 | 52.72±0.45 | 53.82±0.55 | 52.80±0.24 |
| **Architecture-based** | *TrafficStream* | MAE | 18.54±0.53 | 21.49±0.45 | 24.81±0.41 | 21.29±0.47 | 12.94±0.03 | 14.07±0.06 | 16.34±0.08 | 14.23±0.05 | 5.50±0.05 | 5.40±0.19 | 5.40±0.20 | 5.44±0.14 |
| | | RMSE | 28.65±0.70 | 33.98±0.59 | 39.40±0.54 | 33.37±0.63 | 20.83±0.04 | 22.92±0.08 | 26.86±0.11 | 23.15±0.07 | 5.65±0.06 | 5.62±0.14 | 5.70±0.15 | 5.65±0.10 |
| | | MAPE (%) | 23.87±0.21 | 27.80±0.41 | 32.81±0.68 | 27.75±0.42 | 17.89±0.70 | 19.49±0.73 | 23.13±0.73 | 19.83±0.70 | 50.14±1.24 | 50.48±1.65 | 51.84±1.62 | 50.72±1.47 |
| | *ST-LoRA* | MAE | 18.54±0.69 | 21.45±0.66 | 24.65±0.54 | 21.22±0.63 | 12.76±0.05 | 13.88±0.06 | 16.10±0.08 | 14.03±0.05 | 5.34±0.14 | 5.34±0.14 | 5.34±0.15 | 5.38±0.09 |
| | | RMSE | 28.94±1.16 | 34.19±1.12 | 39.40±0.97 | 33.54±1.09 | 20.62±0.08 | 22.68±0.11 | 26.54±0.14 | 22.89±0.09 | 5.59±0.00 | 5.55±0.12 | 5.65±0.13 | 5.59±0.08 |
| | | MAPE (%) | 23.04±0.34 | 26.98±0.31 | 31.90±0.17 | 26.89±0.28 | 17.15±0.24 | 18.59±0.29 | 21.97±0.41 | 18.91±0.29 | 52.60±1.70 | 53.08±1.45 | 54.70±1.35 | 53.34±1.54 |
| | *STKEC* | MAE | 18.87±0.44 | 21.74±0.35 | 24.94±0.17 | 21.52±0.34 | 12.96±0.13 | 14.07±0.11 | 16.33±0.07 | 14.24±0.11 | 5.56±0.12 | 5.57±0.07 | 5.55±0.08 | 5.55±0.09 |
| | | RMSE | 29.92±0.58 | 34.80±0.46 | 39.81±0.22 | 34.25±0.41 | 20.85±0.15 | 22.89±0.12 | 26.80±0.09 | 23.13±0.12 | 5.73±0.10 | 5.78±0.06 | 5.87±0.06 | 5.78±0.08 |
| | | MAPE (%) | 24.12±0.24 | 27.91±0.24 | 32.70±0.14 | 27.83±0.19 | 18.73±0.16 | 20.07±0.43 | 23.30±0.31 | 20.39±0.33 | 53.13±0.16 | 53.74±0.31 | 55.01±0.47 | 53.81±0.30 |
| | *EAC* | MAE | 18.59±0.38 | 21.44±0.30 | 24.63±0.24 | 21.23±0.31 | 12.95±0.31 | 13.85±0.42 | 15.63±0.72 | 13.97±0.46 | 5.20±0.21 | 5.25±0.23 | 5.29±0.19 | 5.24±0.20 |
| | | RMSE | 28.39±0.37 | 33.60±0.24 | 38.85±0.16 | 32.98±0.25 | 20.65±0.43 | 22.33±0.62 | 25.40±1.16 | 22.48±0.69 | 5.45±0.18 | 5.58±0.18 | 5.72±0.13 | 5.57±0.16 |
| | | MAPE (%) | 23.47±0.47 | 27.24±0.43 | 32.07±0.45 | 27.19±0.45 | 19.47±2.64 | 20.39±2.21 | 22.50±2.24 | 20.59±2.25 | 56.19±5.64 | 57.66±5.09 | 58.56±5.34 | 57.38±5.31 |
| | *ST-Adapter* | MAE | 19.11±0.44 | 21.94±0.61 | 25.27±0.77 | 21.77±0.59 | 12.71±0.05 | 13.80±0.05 | 15.97±0.09 | 13.95±0.06 | 5.47±0.06 | 5.37±0.12 | 5.35±0.09 | 5.39±0.09 |
| | | RMSE | 29.14±0.61 | 34.37±0.84 | 39.86±1.03 | 33.81±0.81 | 20.55±0.06 | 22.55±0.07 | 26.31±0.17 | 22.76±0.08 | 5.63±0.06 | 5.59±0.12 | 5.68±0.08 | 5.62±0.10 |
| | | MAPE (%) | 23.65±0.28 | 27.27±0.29 | 31.90±0.36 | 27.22±0.26 | 17.58±0.45 | 18.78±0.31 | 21.71±0.34 | 19.10±0.35 | 51.17±2.42 | 51.59±2.17 | 52.87±2.25 | 51.78±2.20 |
| | *GraphPro* | MAE | 18.92±1.13 | 21.68±0.86 | 24.96±0.71 | 21.53±0.92 | 12.77±0.07 | 13.91±0.09 | 16.20±0.15 | 14.08±0.10 | 5.68±0.14 | 5.50±0.06 | 5.48±0.06 | 5.55±0.06 |
| | | RMSE | 29.68±1.42 | 34.53±0.98 | 39.73±0.74 | 34.04±1.09 | 20.63±0.09 | 22.74±0.13 | 26.68±0.20 | 22.96±0.13 | 5.83±0.14 | 5.72±0.04 | 5.80±0.02 | 5.77±0.06 |
| | | MAPE (%) | 23.56±1.34 | 27.44±1.06 | 32.36±0.78 | 27.36±1.07 | 17.63±1.08 | 19.23±1.14 | 23.04±1.16 | 19.63±1.12 | 53.70±5.22 | 53.67±5.32 | 55.17±5.23 | 54.04±5.34 |
| | *PECPM* | MAE | 18.44±0.18 | 21.36±0.14 | 24.66±0.10 | 21.17±0.15 | 12.75±0.02 | 13.88±0.03 | 16.11±0.06 | 14.03±0.03 | 5.46±0.04 | 5.46±0.04 | 5.48±0.02 | 5.47±0.03 |
| | | RMSE | 28.74±0.22 | 33.89±0.13 | 39.16±0.09 | 33.33±0.16 | 20.61±0.07 | 22.70±0.09 | 26.56±0.15 | 22.91±0.09 | 5.59±0.03 | 5.63±0.03 | 5.74±0.02 | 5.65±0.03 |
| | | MAPE (%) | 23.85±0.85 | 27.73±0.80 | 32.61±0.71 | 27.65±0.79 | 17.63±0.77 | 19.24±0.80 | 22.92±0.85 | 19.60±0.80 | 53.18±2.14 | 53.81±1.93 | 55.31±1.98 | 54.01±2.04 |
| **Regularization-based** | *EWC* | MAE | 18.21±0.44 | 21.19±0.37 | 24.59±0.32 | 21.00±0.38 | 13.05±0.12 | 14.26±0.11 | 16.72±0.10 | 14.45±0.11 | 5.47±0.09 | 5.37±0.14 | 5.37±0.16 | 5.40±0.10 |
| | | RMSE | 28.50±0.39 | 33.85±0.39 | 39.38±0.40 | 33.26±0.39 | 21.14±0.18 | 23.42±0.19 | 27.75±0.22 | 23.69±0.20 | 5.62±0.10 | 5.57±0.11 | 5.67±0.12 | 5.61±0.08 |
| | | MAPE (%) | 23.04±0.77 | 27.07±0.55 | 32.18±0.43 | 27.01±0.59 | 17.32±0.34 | 18.81±0.49 | 22.19±0.71 | 19.13±0.48 | 51.78±0.53 | 52.05±0.94 | 53.43±0.92 | 52.32±0.78 |
| **Replay-based** | *Replay* | MAE | **17.95**±0.27 | 21.06±0.32 | 24.47±0.35 | 20.82±0.31 | 12.96±0.04 | 14.09±0.04 | 16.38±0.07 | 14.27±0.05 | 5.52±0.07 | 5.42±0.21 | 5.42±0.21 | 5.46±0.16 |
| | | RMSE | 28.14±0.46 | 33.57±0.45 | 39.00±0.45 | 32.92±0.45 | 20.84±0.06 | 22.93±0.06 | 26.94±0.10 | 23.19±0.07 | 5.67±0.06 | 5.63±0.18 | 5.72±0.18 | 5.67±0.14 |
| | | MAPE (%) | **22.61**±0.53 | 26.91±0.56 | 32.19±0.68 | 26.77±0.56 | 18.19±0.08 | 19.52±0.11 | 22.66±0.26 | 19.82±0.13 | 52.54±1.52 | 52.95±1.58 | 54.38±1.79 | 53.19±1.62 |
| **Retrieval-based** | STRAP | MAE | 18.04±0.52 | **20.49**±0.41 | **23.45**±0.33 | **20.39**±0.43 | **12.19**±0.18 | **13.13**±0.15 | **15.20**±0.17 | **13.31**±0.17 | **4.83**±0.17 | **4.84**±0.18 | **4.88**±0.17 | **4.85**±0.18 |
| | | RMSE | **26.65**±0.45 | **30.87**±0.39 | **35.56**±0.34 | **30.55**±0.40 | **18.54**±0.25 | **20.18**±0.21 | **23.67**±0.26 | **20.46**±0.23 | **4.95**±0.18 | **5.01**±0.19 | **5.15**±0.17 | **5.03**±0.18 |
| | | MAPE (%) | 23.72±0.59 | **26.78**±0.43 | **30.73**±0.37 | **26.74**±0.43 | **16.55**±0.55 | **17.70**±0.69 | **20.57**±0.97 | **18.02**±0.72 | **42.18**±1.64 | **43.02**±1.77 | **44.30**±1.55 | **43.11**±1.72 |

## 5.3 Case Study (RQ4)

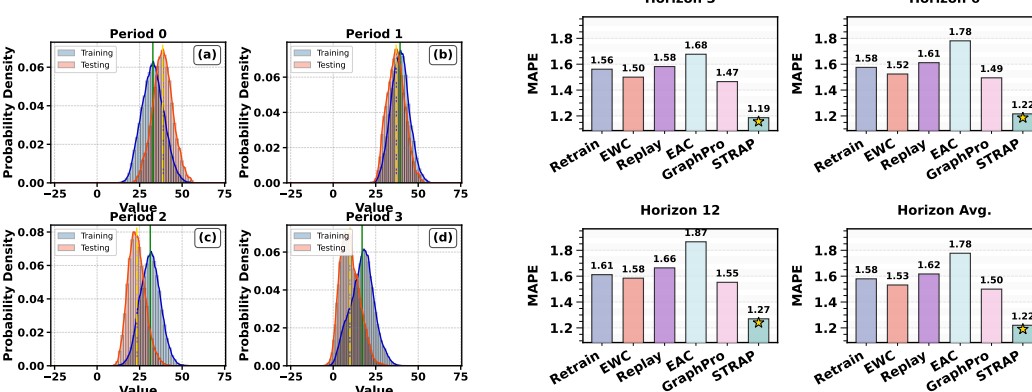

Figure 3: Test set distributions for ENERGY-Wind for 4 periods 0 (a), 1 (b), 2 (c), and 3 (d).

Figure 4: Performance analysis (MAPE) of different horizons across various baselines.

To further investigate the effectiveness of our proposed STRAP method under significant distribution shifts, we conduct a detailed case study on the ENERGY-Wind dataset. As illustrated in Figure 3, the distributions of test set values vary considerably across the four periods, with period 3 (Figure 4(d)) exhibiting the most pronounced shift from the initial distribution (Figure 4(a)). This is evidenced by

the substantial decrease in both mean and median values, representing a reduction of approximately 77% and 76% respectively.

Figure 4 presents the performance comparison across different prediction horizons during period 3 on STGNN backbone (More experiments are presented in C.5 in Appendix C), which experiences the most severe distribution shift. **The results demonstrate that STRAP significantly outperforms all baseline models across all prediction horizons**, achieving the lowest MAPE of 1.19, 1.22, 1.27, and 1.22 for horizons 3, 6, 12, and on average, respectively. This remarkable performance can be attributed to STRAP's retrieval-based key-value mechanism, which selectively identifies and extracts the most information-rich patterns from historical knowledge while efficiently integrating them with current observations, thus achieving an optimal balance that enables robust prediction even under extreme distribution shifts.

### 5.4 Few-shot Learning Performance Evaluation (RQ6)

We evaluate STRAP under data-constrained conditions on the Energy-Wind dataset with 50% and 30% missing data scenarios (Table 2). STRAP consistently outperforms all baseline methods across both settings. Under severe constraints (50% missing data), STRAP achieves 6.29 MAE, outperforming the best baseline EWC by 8.7%. With moderate constraints (30% missing data), STRAP maintains superiority with 5.77 MAE, surpassing Graphpro by 3.4%. The method demonstrates exceptional relative accuracy with MAPE scores of 78.92% and 67.42% respectively, while maintaining consistent performance across different prediction horizons.

Table 2: Few-shot Performance Comparison on TGCN Backbone (Energy-Stream).

| Data Availability | | | 50% Missing Data | | | | 30% Missing Data | | | | All Data | | | |
|---|---|---|---|---|---|---|---|---|---|---|---|---|---|---|
| Category | Method | Metric | 3 | 6 | 12 | Avg. | 3 | 6 | 12 | Avg. | 3 | 6 | 12 | Avg. |
| Back-bone | Pretrain | MAE | 7.16±0.12 | 7.16±0.11 | 7.15±0.09 | 7.16±0.11 | 6.23±0.08 | 6.17±0.07 | 6.21±0.06 | 6.18±0.07 | 8.94±0.15 | 8.92±0.14 | 8.93±0.13 | 8.93±0.14 |
| | | RMSE | 7.28±0.13 | 7.31±0.12 | 7.37±0.10 | 7.32±0.12 | 6.41±0.09 | 6.40±0.08 | 6.53±0.07 | 6.41±0.08 | 9.12±0.16 | 9.15±0.15 | 9.18±0.14 | 9.15±0.15 |
| | | MAPE (%) | 92.44±2.18 | 93.38±1.95 | 94.65±1.82 | 93.36±1.98 | 75.08±1.85 | 75.47±1.72 | 76.77±1.68 | 75.60±1.75 | 118.25±3.42 | 119.15±3.28 | 120.38±3.15 | 119.26±3.28 |
| | Retrain | MAE | 6.91±0.08 | 6.89±0.07 | 6.86±0.06 | 6.89±0.07 | 6.35±0.05 | 6.28±0.04 | 6.27±0.03 | 6.27±0.04 | 5.68±0.12 | 5.50±0.08 | 5.48±0.06 | 5.55±0.09 |
| | | RMSE | 7.05±0.09 | 7.06±0.08 | 7.10±0.07 | 7.06±0.08 | 6.55±0.06 | 6.52±0.05 | 6.61±0.04 | 6.53±0.05 | 5.83±0.14 | 5.72±0.04 | 5.80±0.02 | 5.77±0.06 |
| | | MAPE (%) | 89.03±1.95 | 89.91±1.82 | 90.96±1.75 | 89.78±1.84 | 70.70±1.25 | 71.28±1.18 | 72.51±1.12 | 71.32±1.18 | 53.70±2.15 | 53.67±1.98 | 55.17±1.85 | 54.04±1.99 |
| Architecture-based | Graphpro | MAE | 6.94±0.15 | 6.92±0.12 | 6.93±0.11 | 6.93±0.13 | 6.07±0.08 | 5.96±0.06 | 6.00±0.05 | 5.97±0.06 | 5.68±0.14 | 5.50±0.06 | 5.48±0.06 | 5.55±0.06 |
| | | RMSE | 7.04±0.16 | 7.06±0.13 | 7.14±0.12 | 7.07±0.14 | 6.28±0.09 | 6.20±0.07 | 6.34±0.06 | 6.23±0.07 | 5.83±0.14 | 5.72±0.04 | 5.80±0.02 | 5.77±0.06 |
| | | MAPE (%) | 91.25±2.85 | 92.12±2.65 | 93.69±2.48 | 92.18±2.66 | 74.31±1.95 | 74.73±1.82 | 76.14±1.75 | 74.86±1.84 | 53.70±5.22 | 53.67±5.32 | 55.17±5.23 | 54.04±5.34 |
| | ST-Adapter | MAE | 7.02±0.18 | 7.01±0.16 | 7.02±0.15 | 7.03±0.16 | 6.19±0.12 | 6.16±0.10 | 6.22±0.09 | 6.16±0.10 | 5.47±0.06 | 5.37±0.12 | 5.35±0.09 | 5.39±0.09 |
| | | RMSE | 7.24±0.19 | 7.28±0.17 | 7.38±0.16 | 7.30±0.17 | 6.48±0.13 | 6.47±0.11 | 6.61±0.10 | 6.48±0.11 | 5.63±0.06 | 5.59±0.12 | 5.68±0.08 | 5.62±0.10 |
| | | MAPE (%) | 98.47±3.25 | 99.12±3.08 | 99.98±2.95 | 99.09±3.09 | 76.15±2.15 | 76.64±2.05 | 77.96±1.98 | 76.74±2.06 | 51.17±2.42 | 51.59±2.17 | 52.87±2.25 | 51.78±2.20 |
| | EWC | MAE | 6.91±0.12 | 6.89±0.10 | 6.86±0.09 | 6.89±0.10 | 6.35±0.08 | 6.28±0.06 | 6.27±0.05 | 6.27±0.06 | 5.47±0.09 | 5.37±0.14 | 5.37±0.16 | 5.40±0.10 |
| | | RMSE | 7.05±0.13 | 7.06±0.11 | 7.10±0.10 | 7.06±0.11 | 6.55±0.09 | 6.52±0.07 | 6.61±0.04 | 6.53±0.07 | 5.62±0.10 | 5.57±0.11 | 5.67±0.12 | 5.61±0.08 |
| | | MAPE (%) | 89.03±2.25 | 89.91±2.12 | 90.96±2.05 | 89.78±2.14 | 70.70±1.85 | 71.28±1.75 | 72.51±1.68 | 71.32±1.76 | 51.78±0.53 | 52.05±0.94 | 53.43±0.92 | 52.32±0.78 |
| Replay-based | Replay | MAE | 7.16±0.15 | 7.16±0.14 | 7.15±0.13 | 7.16±0.14 | 6.23±0.10 | 6.17±0.08 | 6.21±0.07 | 6.18±0.08 | 5.52±0.07 | 5.42±0.21 | 5.42±0.21 | 5.46±0.16 |
| | | RMSE | 7.28±0.16 | 7.31±0.15 | 7.37±0.14 | 7.32±0.15 | 6.41±0.11 | 6.40±0.08 | 6.53±0.06 | 6.41±0.09 | 5.67±0.06 | 5.63±0.18 | 5.72±0.18 | 5.67±0.14 |
| | | MAPE (%) | 92.44±2.95 | 93.38±2.82 | 94.65±2.75 | 93.36±2.84 | 75.08±2.05 | 75.47±1.95 | 76.77±1.88 | 75.60±1.96 | 52.54±1.52 | 52.95±1.58 | 54.38±1.79 | 53.19±1.62 |
| Retrieval-based | STRAP | MAE | **6.31±0.08** | **6.29±0.07** | **6.28±0.06** | **6.29±0.07** | **5.74±0.05** | **5.64±0.04** | **5.66±0.03** | **5.77±0.04** | **4.83±0.17** | **4.84±0.18** | **4.88±0.17** | **4.85±0.18** |
| | | RMSE | **6.44±0.09** | **6.46±0.08** | **6.53±0.07** | **6.47±0.08** | **6.10±0.06** | **6.01±0.05** | **6.11±0.04** | **6.14±0.05** | **4.95±0.18** | **5.01±0.19** | **5.15±0.17** | **5.03±0.18** |
| | | MAPE (%) | **78.28±1.85** | **78.93±1.75** | **79.98±1.68** | **78.92±1.76** | **66.53±1.25** | **67.02±1.18** | **68.36±1.12** | **67.42±1.18** | **42.18±1.64** | **43.02±1.77** | **44.30±1.55** | **43.11±1.72** |

## 6 Conclusion

We introduced STRAP, a novel framework for streaming graph prediction that leverages specialized pattern libraries across spatial, temporal, and spatiotemporal dimensions to capture the complex dynamics of evolving graphs. It encompasses three innovative components: spatial temporal subgraph chunking, multi-dimensional pattern library construction, and pattern retrieval and knowledge fusion. Extensive experiments show that STRAP consistently surpasses state-of-the-art methods across various backbone architectures. However, our approach faces limitations: rising computational demands with larger libraries and reliance on robust historical datasets. In the future, we aim to develop adaptive library management techniques, integrate uncertainty measures, and expand STRAP to multi-modal graph scenarios.

## 7 Acknowledgements

The paper is supported by the Young Scientists Fund Program C of the National Natural Science Foundation of China (Grant No. 62502406) and the Start-up Fund from the City University of Hong Kong (Dongguan).

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

## A Notations

The notations in this paper are summarized in Table 3.

Table 3: Notations Tables in STRAP

| Notation | Definition |
|----------|------------|
| $\mathcal{G}_t$ / $V_t$ / $A_t$ | The dynamic graph / node set / adjacency matrix at time $t$ |
| $X_t$ | The feature matrix at time $t$ |
| $e^{tr}$ / $e^{te}$ | The training environment / test environment |
| $\mathcal{C}$ | The set of graph communities |
| $\mathcal{N}(v)$ | The neighbors of node $v$ |
| $N_k(v_c)$ | $k$-hop neighborhood of central node $v_c$ |
| $\mathcal{G}_S$ | The spatial subgraph |
| $\mathcal{G}_T$ | The temporal subgraph |
| $\mathcal{G}_{ST}$ | The spatio-temporal subgraph |
| $\tau_S$ / $\tau_{ST}$ | Time length of spatial / spatio-temporal windows |
| $\mathcal{B}^S$ / $\mathcal{B}^T$ / $\mathcal{B}^{ST}$ | The spatial / temporal / spatio-temporal pattern library |
| $k^S$ / $k^T$ / $k^{ST}$ | The spatial / temporal / spatio-temporal pattern keys |
| $v^S$ / $v^T$ / $v^{ST}$ | The spatial / temporal / spatio-temporal pattern values |
| $D(\mathcal{G}_\mathcal{S})$ | The neighborhood statistics of graph $\mathcal{G}_\mathcal{S}$ |
| $CL(\mathcal{G}_\mathcal{S})$ | The clustering coefficients of graph $\mathcal{G}_\mathcal{S}$ |
| $SP(\mathcal{G}_\mathcal{S})$ | The shortest path statistics of graph $\mathcal{G}_\mathcal{S}$ |
| $FR(v_i)$ | The Forman-Ricci curvature of node $v_i$ |
| $S(X)$ | The statistical moments of temporal signal $X$ |
| $F(X)$ | The frequency components of temporal signal $X$ |
| $W(X)$ | The wavelet transform of temporal signal $X$ |
| $R(X)$ | The autocorrelation functions of temporal signal $X$ |
| $H(X)$ | The entropy of temporal signal $X$ |
| $s(k_i^q, k_j)$ | The similarity between query key $k_i^q$ and stored key $k_j$ |
| $\mathcal{R}_i$ | The retrieved key-value pairs for query $i$ |
| $v_i^*$ | The similarity-weighted average of retrieved values |
| $\Theta_{pt}$ / $\Theta_{train}$ | The pretrained / trainable backbone parameters |
| $Z_1$ / $Z_2$ | The current observation / historical pattern embeddings |
| $\gamma$ | The balance weight between current and historical information |
| $\mathcal{L}$ | The loss function for prediction |
| $T$ | The number of time steps |
| $T_p$ | The prediction horizon |
| $N$ | The number of nodes |
| $c$ | The dimension of node features |
| $\otimes$ | Element-wise cross-product operation |

## B Further Methods Details

### B.1 Theoretical Analysis

**Theorem B.1.** *Let $X$ be the original data. Consider two feature extraction approaches:*

- *Decomposed approach: Features from separate libraries $(f_S(X), f_T(X), f_{ST}(X))$*

- *Unified approach: Features from a parametric model $f_\theta(X)$*

*The mutual information advantage of the decomposed approach satisfies:*

$$I(X; f_S(X), f_T(X), f_{ST}(X)) - I(X; f_\theta(X)) > 0 \tag{13}$$

*Proof.* Let us denote: $S = f_S(X), T = f_T(X), ST = f_{ST}(X), Z = f_\theta(X)$.

We need to prove:

$$I(X;S,T,ST) - I(X;Z) > 0 \tag{14}$$

By definition of mutual information:

$$I(X;S,T,ST) - I(X;Z)$$

$$= \sum_{X,S,T,ST} p(X,S,T,ST) \log \frac{p(X,S,T,ST)}{p(X)p(S,T,ST)} - \sum_{X,Z} p(X,Z) \log \frac{p(X,Z)}{p(X)p(Z)} \tag{15}$$

$$= \sum_{X,S,T,ST} p(X,S,T,ST) \log \frac{p(X|S,T,ST)}{p(X)} - \sum_{X,Z} p(X,Z) \log \frac{p(X|Z)}{p(X)} \tag{16}$$

We can rewrite the second term by marginalizing over $(S,T,ST)$:

$$\sum_{X,Z} p(X,Z) \log \frac{p(X|Z)}{p(X)} = \sum_{X,S,T,ST,Z} p(X,S,T,ST,Z) \log \frac{p(X|Z)}{p(X)} \tag{15}$$

$$= \sum_{X,S,T,ST,Z} p(Z|X,S,T,ST) p(X,S,T,ST) \log \frac{p(X|Z)}{p(X)} \tag{16}$$

This gives us:

$$I(X;S,T,ST) - I(X;Z) \tag{17}$$

$$= \sum_{X,S,T,ST} p(X,S,T,ST) \log \frac{p(X|S,T,ST)}{p(X)} - \sum_{X,S,T,ST,Z} p(Z|X,S,T,ST) p(X,S,T,ST) \log \frac{p(X|Z)}{p(X)} \tag{18}$$

$$= \sum_{X,S,T,ST} p(X,S,T,ST) \left[ \log \frac{p(X|S,T,ST)}{p(X)} - \sum_{Z} p(Z|X,S,T,ST) \log \frac{p(X|Z)}{p(X)} \right] \tag{19}$$

$$= \sum_{X,S,T,ST} p(X,S,T,ST) \left[ \log p(X|S,T,ST) - \log p(X) - \sum_{Z} p(Z|X,S,T,ST) \log \frac{p(X|Z)}{p(X)} \right] \tag{20}$$

$$= \sum_{X,S,T,ST} p(X,S,T,ST) \left[ \log p(X|S,T,ST) - \log p(X) - \sum_{Z} p(Z|X,S,T,ST) \log p(X|Z) \right. \tag{21}$$

$$\left. + \sum_{Z} p(Z|X,S,T,ST) \log p(X) \right] \tag{22}$$

Since $\sum_{Z} p(Z|X,S,T,ST) = 1$, we have:

$$I(X;S,T,ST) - I(X;Z) \tag{23}$$

$$= \sum_{X,S,T,ST} p(X,S,T,ST) [\log p(X|S,T,ST) \tag{24}$$

$$- \log p(X) - \sum_{Z} p(Z|X,S,T,ST) \log p(X|Z) + \log p(X) \Big] \tag{25}$$

$$= \sum_{X,S,T,ST} p(X,S,T,ST) \left[ \log p(X|S,T,ST) - \sum_{Z} p(Z|X,S,T,ST) \log p(X|Z) \right] \tag{26}$$

Using Jensen's inequality for the concave function $\log$:

$$\sum_{Z} p(Z|X,S,T,ST) \log p(X|Z) \leq \log \sum_{Z} p(Z|X,S,T,ST) p(X|Z) \tag{27}$$

Therefore:

$$I(X;S,T,ST) - I(X;Z) \geq \sum_{X,S,T,ST} p(X,S,T,ST)[\log p(X|S,T,ST) \tag{28}$$

$$-\log \sum_Z p(Z|X,S,T,ST)p(X|Z)\bigg] \tag{29}$$

Due to the Markov chain $Z - X - (S,T,ST)$, we have $p(Z|X,S,T,ST) = p(Z|X)$. This Markov property holds because all features $S$, $T$, $ST$ and $Z$ are deterministic functions of $X$, making them conditionally independent given $X$.

Therefore:

$$I(X;S,T,ST) - I(X;Z) \geq \sum_{X,S,T,ST} p(X,S,T,ST)[\log p(X|S,T,ST) \tag{30}$$

$$-\log \sum_Z p(Z|X)p(X|Z)\bigg] \tag{31}$$

Since $Z = f_\theta(X)$ is a deterministic function, $p(Z|X)$ is a point mass at $Z_X = f_\theta(X)$, giving:

$$\sum_Z p(Z|X)p(X|Z) = p(X|Z_X) \tag{32}$$

Similarly, for the deterministic functions $S_X = f_S(X)$, $T_X = f_T(X)$, and $ST_X = f_{ST}(X)$:

$$p(X|S,T,ST) = p(X|S_X,T_X,ST_X) \tag{33}$$

By Bayes' theorem:

$$p(X|S_X,T_X,ST_X) = \frac{p(X)}{p(S_X,T_X,ST_X)} \tag{34}$$

$$p(X|Z_X) = \frac{p(X)}{p(Z_X)} \tag{35}$$

For any given $X$:

$$\{X': f_S(X') = S_X, f_T(X') = T_X, f_{ST}(X') = ST_X\} \subseteq \{X': f_\theta(X') = Z_X\} \tag{36}$$

This directly implies:

$$p(S_X,T_X,ST_X) = \sum_{X': f_S(X')=S_X, f_T(X')=T_X, f_{ST}(X')=ST_X} p(X') \tag{37}$$

$$< \sum_{X': f_\theta(X')=Z_X} p(X') \tag{38}$$

$$= p(Z_X) \tag{39}$$

Therefore:

$$p(X|S_X,T_X,ST_X) = \frac{p(X)}{p(S_X,T_X,ST_X)} > \frac{p(X)}{p(Z_X)} = p(X|Z_X) \tag{40}$$

Taking logarithms:

$$\log p(X|S,T,ST) > \log p(X|Z_X) = \log \sum_Z p(Z|X)p(X|Z) \tag{41}$$

Substituting back:

$$I(X;S,T,ST) - I(X;Z) > 0 \tag{42}$$

Therefore:

$$I(X;f_S(X),f_T(X),f_{ST}(X)) - I(X;f_\theta(X)) > 0 \tag{43}$$

$\square$

## B.2 Algorithms

Algorithm 1 presents the Spatio-Temporal Pattern Library Construction process, which forms the foundation of our retrieval-augmented framework. Given a dynamic spatio-temporal graph, this algorithm systematically extracts and stores multi-dimensional patterns across spatial, temporal, and spatio-temporal domains. For each snapshot, we decompose the graph into specialized subgraphs using domain-specific chunking methods, then generate discriminative keys and informative values for each pattern type. Keys are extracted using topological properties for spatial patterns, time series characteristics for temporal patterns, and cross-dimensional interactions for spatio-temporal patterns, while values are generated by applying a pre-trained backbone to preserve essential features. The algorithm also implements an importance-based sampling strategy to manage historical patterns efficiently, ensuring the libraries maintain the most informative patterns while controlling memory usage.

Algorithm 2 outlines the Retrieval-Augmented Spatio-Temporal Prediction framework that leverages the previously constructed pattern libraries to enhance prediction performance under distribution shifts. When presented with new input features and graph structure, the algorithm first projects the query features into the pattern space and performs multi-dimensional retrieval across spatial, temporal, and spatio-temporal libraries based on similarity scores. For each pattern type, it retrieves the top-k most relevant patterns and computes similarity-weighted averages of their values. The algorithm then implements an adaptive knowledge fusion mechanism that balances the contributions from current observations (processed by the backbone) and retrieved historical knowledge through a learnable fusion weight, enabling the model to dynamically adjust the influence of historical patterns based on their relevance to the current input. This fused representation is finally passed to a decoder to generate the enhanced prediction.

# C  Additional Experiment Details

Table 4: Comparison of the overall performance of different methods (TGCN backbone).

| Datasets | | | Air-Stream | | | | PEMS-Stream | | | | Energy-Stream | | | |
|---|---|---|---|---|---|---|---|---|---|---|---|---|---|---|
| Category | Method | Metric | 3 | 6 | 12 | Avg. | 3 | 6 | 12 | Avg. | 3 | 6 | 12 | Avg. |
| Back-bone | Pretrain | MAE | 18.61±0.35 | 21.40±0.25 | 24.60±0.17 | 21.23±0.25 | 15.19±0.53 | 15.86±0.29 | 17.98±0.31 | 16.15±0.38 | 10.63±0.01 | 10.63±0.01 | 10.64±0.01 | 10.63±0.01 |
| | | RMSE | 29.51±0.78 | 34.26±0.60 | 39.28±0.41 | 33.79±0.62 | 23.59±0.91 | 24.93±0.54 | 28.62±0.55 | 25.37±0.68 | 10.74±0.00 | 10.77±0.00 | 10.84±0.02 | 10.78±0.01 |
| | | MAPE(%) | 23.21±0.50 | 27.37±0.58 | 32.51±0.66 | 27.23±0.59 | 31.98±3.28 | 32.47±3.10 | 35.17±2.84 | 32.97±3.07 | 169.64±0.32 | 170.30±0.34 | 171.53±0.40 | 170.39±0.33 |
| | Retrain | MAE | 18.75±0.20 | 21.52±0.12 | 24.68±0.07 | 21.34±0.13 | 13.21±0.10 | 14.19±0.07 | 16.41±0.05 | 14.39±0.07 | 5.50±0.07 | 5.48±0.08 | 5.50±0.08 | 5.49±0.08 |
| | | RMSE | 29.22±0.25 | 34.12±0.16 | 39.20±0.08 | 33.61±0.17 | 21.40±0.16 | 23.19±0.11 | 27.01±0.10 | 23.50±0.13 | 5.63±0.06 | 5.66±0.06 | 5.78±0.06 | 5.69±0.06 |
| | | MAPE(%) | 23.40±0.36 | 27.29±0.45 | 32.20±0.55 | 27.21±0.43 | 18.78±0.60 | 19.83±0.48 | 22.46±0.27 | 20.12±0.45 | 54.65±2.46 | 55.00±2.01 | 56.39±2.04 | 55.18±2.12 |
| Architecture-Based | TrafficStream | MAE | 18.08±0.23 | 21.03±0.24 | 24.37±0.25 | 20.83±0.23 | 13.40±0.17 | 14.42±0.14 | 16.67±0.14 | 14.62±0.15 | 5.45±0.10 | 5.44±0.12 | 5.44±0.13 | 5.44±0.11 |
| | | RMSE | 28.86±0.71 | 33.91±0.57 | 39.11±0.46 | 33.36±0.59 | 21.77±0.17 | 23.63±0.13 | 27.46±0.11 | 23.91±0.14 | 5.58±0.09 | 5.61±0.11 | 5.70±0.12 | 5.62±0.11 |
| | | MAPE(%) | 22.80±0.84 | 27.03±0.74 | 32.24±0.62 | 26.91±0.74 | 19.28±0.12 | 20.63±0.16 | 24.04±0.41 | 20.99±0.16 | 52.02±0.22 | 52.43±0.25 | 53.85±0.25 | 52.63±0.11 |
| | ST-LoRA | MAE | 18.41±0.07 | 21.26±0.08 | 24.50±0.11 | 21.08±0.08 | 13.06±0.12 | 14.08±0.09 | 16.30±0.08 | 14.27±0.09 | 5.40±0.10 | 5.40±0.10 | 5.41±0.09 | 5.40±0.10 |
| | | RMSE | 28.74±0.31 | 33.81±0.22 | 39.02±0.15 | 33.26±0.24 | 21.08±0.20 | 22.97±0.16 | 26.80±0.13 | 23.24±0.17 | 5.54±0.09 | 5.59±0.09 | 5.69±0.08 | 5.60±0.09 |
| | | MAPE(%) | 23.09±0.41 | 27.03±0.37 | 31.97±0.36 | 26.94±0.38 | 18.86±0.48 | 20.11±0.50 | 23.14±0.84 | 20.40±0.59 | 53.53±3.83 | 54.13±3.82 | 55.42±3.71 | 54.24±3.77 |
| | STKEC | MAE | 18.75±0.99 | 21.52±0.73 | 24.65±0.47 | 21.34±0.77 | 13.49±0.18 | 14.52±0.12 | 16.74±0.10 | 14.72±0.13 | 5.34±0.02 | 5.33±0.06 | 5.35±0.09 | 5.33±0.03 |
| | | RMSE | 29.31±0.90 | 34.27±0.66 | 39.35±0.49 | 33.73±0.72 | 22.04±0.28 | 23.88±0.16 | 27.67±0.13 | 24.17±0.19 | 5.48±0.03 | 5.52±0.05 | 5.64±0.08 | 5.53±0.02 |
| | | MAPE(%) | 23.71±1.37 | 27.60±0.91 | 32.54±0.54 | 27.54±0.98 | 18.94±0.30 | 20.25±0.37 | 23.33±0.52 | 20.55±0.39 | 54.15±0.48 | 54.12±1.11 | 55.31±1.42 | 54.40±0.62 |
| | EAC | MAE | 19.21±1.31 | 21.83±1.11 | 24.82±0.89 | 21.67±1.13 | 12.69±0.09 | 13.42±0.10 | 14.83±0.11 | 13.51±0.10 | 5.23±0.19 | 5.24±0.20 | 5.27±0.20 | 5.25±0.20 |
| | | RMSE | 29.74±1.84 | 34.48±1.61 | 39.37±1.32 | 33.97±1.63 | 20.21±0.13 | 21.56±0.13 | 23.92±0.15 | 21.66±0.13 | 5.38±0.19 | 5.45±0.19 | 5.58±0.20 | 5.46±0.19 |
| | | MAPE(%) | 23.73±1.20 | 27.32±0.82 | 31.87±0.45 | 27.27±0.87 | 18.72±0.58 | 19.40±0.53 | 20.95±0.58 | 19.52±0.55 | 50.84±3.13 | 51.66±3.16 | 53.15±3.19 | 51.79±3.14 |
| | ST-Adapter | MAE | 18.12±0.24 | 21.08±0.20 | 24.38±0.18 | 20.86±0.20 | 13.04±0.03 | 14.07±0.01 | 16.22±0.02 | 14.24±0.01 | 5.32±0.06 | 5.31±0.05 | 5.31±0.04 | 5.32±0.05 |
| | | RMSE | 28.32±0.29 | 33.54±0.19 | 38.84±0.07 | 32.95±0.20 | 20.94±0.06 | 22.84±0.01 | 26.57±0.04 | 23.08±0.02 | 5.46±0.05 | 5.49±0.05 | 5.58±0.04 | 5.50±0.05 |
| | | MAPE(%) | 23.06±0.74 | 27.02±0.57 | 31.98±0.47 | 26.92±0.61 | 20.03±0.09 | 21.23±0.09 | 23.99±0.11 | 21.49±0.08 | 52.36±0.97 | 53.18±1.19 | 54.67±1.46 | 53.28±1.20 |
| | GraphPro | MAE | 18.41±0.13 | 21.20±0.09 | 24.40±0.08 | 21.03±0.10 | 12.90±0.08 | 13.97±0.07 | 16.17±0.07 | 14.14±0.07 | 5.47±0.10 | 5.52±0.04 | 5.54±0.01 | 5.51±0.05 |
| | | RMSE | 28.65±0.14 | 33.68±0.15 | 38.86±0.14 | 33.13±0.15 | 20.92±0.16 | 22.87±0.13 | 26.65±0.12 | 23.11±0.14 | 5.63±0.08 | 5.72±0.05 | 5.84±0.03 | 5.72±0.05 |
| | | MAPE(%) | 22.88±0.09 | 26.78±0.11 | 31.77±0.14 | 26.73±0.11 | 18.86±0.84 | 19.96±0.75 | 22.69±0.56 | 20.25±0.73 | 52.39±2.27 | 53.15±2.10 | 54.55±1.89 | 53.24±2.09 |
| | PECPM | MAE | 18.90±0.39 | 21.70±0.39 | 24.88±0.39 | 21.51±0.39 | 13.58±0.77 | 14.54±0.67 | 16.67±0.61 | 14.73±0.69 | 5.53±0.07 | 5.52±0.08 | 5.53±0.08 | 5.53±0.07 |
| | | RMSE | 29.70±0.64 | 34.61±0.52 | 39.61±0.41 | 34.06±0.55 | 22.61±2.20 | 24.35±1.95 | 27.90±1.64 | 24.61±1.96 | 5.72±0.11 | 5.76±0.11 | 5.86±0.12 | 5.78±0.12 |
| | | MAPE(%) | 23.51±1.10 | 27.40±0.97 | 32.33±0.83 | 27.33±0.97 | 19.16±0.12 | 20.26±0.13 | 23.11±0.34 | 20.56±0.15 | 51.13±4.72 | 51.59±4.53 | 52.88±4.20 | 51.77±4.52 |
| Regularization-based | EWC | MAE | 18.90±0.46 | 21.71±0.40 | 24.90±0.33 | 21.53±0.40 | 13.83±0.08 | 14.86±0.05 | 17.16±0.06 | 15.07±0.06 | 5.40±0.09 | 5.40±0.09 | 5.41±0.09 | 5.40±0.09 |
| | | RMSE | 29.75±1.02 | 34.67±0.80 | 39.77±0.61 | 34.15±0.84 | 22.96±0.19 | 24.81±0.15 | 28.70±0.19 | 25.12±0.17 | 5.54±0.08 | 5.58±0.08 | 5.69±0.07 | 5.60±0.08 |
| | | MAPE(%) | 23.68±0.55 | 27.63±0.43 | 32.55±0.31 | 27.54±0.42 | 19.04±0.57 | 20.22±0.59 | 23.28±0.68 | 20.56±0.61 | 49.59±1.50 | 50.20±1.56 | 51.51±1.63 | 50.31±1.56 |
| Replay-based | Replay | MAE | 17.97±0.28 | 20.91±0.24 | 24.22±0.18 | 20.71±0.23 | 13.61±0.09 | 14.66±0.08 | 16.98±0.11 | 14.88±0.09 | 5.43±0.02 | 5.42±0.03 | 5.41±0.05 | 5.42±0.03 |
| | | RMSE | 28.63±0.75 | 33.69±0.56 | 38.89±0.40 | 33.14±0.60 | 22.05±0.17 | 23.91±0.14 | 27.85±0.17 | 24.24±0.16 | 5.57±0.02 | 5.60±0.03 | 5.69±0.05 | 5.61±0.04 |
| | | MAPE(%) | 22.77±0.39 | 26.90±0.26 | 32.06±0.12 | 26.80±0.27 | 20.53±0.11 | 21.58±0.37 | 24.39±0.49 | 21.90±0.30 | 49.88±1.97 | 50.45±1.99 | 51.64±2.12 | 50.54±2.03 |
| Retrieval-based | STRAP | MAE | 18.59±0.62 | 20.91±0.60 | 23.69±0.53 | 20.81±0.59 | 12.31±0.02 | 13.21±0.04 | 15.23±0.04 | 13.39±0.03 | 4.96±0.10 | 4.97±0.11 | 5.00±0.10 | 4.98±0.10 |
| | | RMSE | 27.66±1.06 | 31.67±0.92 | 36.11±0.81 | 31.36±0.94 | 18.81±0.05 | 20.36±0.07 | 23.76±0.07 | 20.66±0.07 | 5.17±0.08 | 5.22±0.09 | 5.34±0.08 | 5.23±0.08 |
| | | MAPE(%) | 23.53±0.59 | 26.52±0.45 | 30.36±0.30 | 26.48±0.44 | 16.94±1.02 | 17.98±0.83 | 20.87±0.38 | 18.33±0.78 | 42.83±3.85 | 43.53±3.72 | 44.89±3.50 | 43.65±3.73 |

## C.1  Implementation Details and Evaluation

We establish a data split ratio of training/validation/test = 6/2/2 for all experiments. For fair comparisons, we set the learning rate to either 0.03 or 0.01 based on the specific situation of each dataset and model requirements (Table 7). The parameters of baselines were set based on their original papers and any

---

**Algorithm 1** Spatio-Temporal Pattern Library Construction

---

**Require:** Dynamic spatio-temporal graph $\mathcal{G}$, feature dimension $d$, pre-trained backbone $f_\theta$, history ratio $\beta$, retrieval count $k$

**Ensure:** Spatio-temporal pattern libraries $\mathcal{B}^S, \mathcal{B}^T, \mathcal{B}^{ST}$

 1: Initialize pattern libraries $\mathcal{B}^S \leftarrow \emptyset, \mathcal{B}^T \leftarrow \emptyset, \mathcal{B}^{ST} \leftarrow \emptyset$

 2: **for** each snapshot $G_\tau \in \mathcal{G}$ **do**                                         ▷ **Extract Patterns**

 3:     **Spatial Patterns:**

 4:     $\{G^S\} \leftarrow \text{SPATIALSUBGRAPHCHUNKING}(G_\tau, \tau_S)$                    ▷ Via Eq. (2)

 5:     **for** each subgraph $G_i^S \in \{G^S\}$ **do**

 6:         Extract key $k_i^S \leftarrow \text{EXTRACTSPATIALKEY}(G_i^S)$                     ▷ Via Eq. (5)

 7:         Extract value $v_i^S \leftarrow \text{MEANPOOL}(f_\theta(G_i^S))$                   ▷ Via Eq. (8)

 8:         Store pattern $(k_i^S, v_i^S)$ in $\mathcal{B}^S$

 9:     **end for**

10:     **Temporal Patterns:**

11:     $\{G^T\} \leftarrow \text{TEMPORALSUBGRAPHCHUNKING}(G_\tau, k\text{-hop})$            ▷ Via Eq. (3)

12:     **for** each subgraph $G_i^T \in \{G^T\}$ **do**

13:         Extract key $k_i^T \leftarrow \text{EXTRACTTEMPORALKEY}(G_i^T)$                    ▷ Via Eq. (6)

14:         Extract value $v_i^T \leftarrow \text{MEANPOOL}(f_\theta(G_i^T))$                   ▷ Via Eq. (8)

15:         Store pattern $(k_i^T, v_i^T)$ in $\mathcal{B}^T$

16:     **end for**

17:     **Spatio-Temporal Patterns:**

18:     $\{G^{ST}\} \leftarrow \text{SPECTRALCLUSTERING}(G_\tau, \tau_{ST})$                  ▷ Via Eq. (4)

19:     **for** each subgraph $G_i^{ST} \in \{G^{ST}\}$ **do**

20:         Extract key $k_i^{ST} \leftarrow k_i^S \otimes k_i^T$                             ▷ Via Eq. (7)

21:         Extract value $v_i^{ST} \leftarrow \text{MEANPOOL}(f_\theta(G_i^{ST}))$           ▷ Via Eq. (8)

22:         Store pattern $(k_i^{ST}, v_i^{ST})$ in $\mathcal{B}^{ST}$

23:     **end for**

24:     **Historical Patterns Management:**

25:     **for** each library $\mathcal{B} \in \{\mathcal{B}^S, \mathcal{B}^T, \mathcal{B}^{ST}\}$ **do**

26:         **if** $|\mathcal{B}| > \text{max\_history\_size}$ **then**

27:             Calculate pattern importance $I(k_i) \leftarrow \frac{1}{\|k_i\| + \epsilon}$

28:             Sample indices $\mathcal{I} \leftarrow \text{WEIGHTEDSAMPLING}(\mathcal{B}, I, \beta \cdot |\mathcal{B}|)$

29:             $\mathcal{B} \leftarrow \{(k_i, v_i) | i \in \mathcal{I}\}$

30:         **end if**

31:         Build search index for $\mathcal{B}$

32:     **end for**

33: **end for**

34: **return** Pattern libraries $\mathcal{B}^S, \mathcal{B}^T, \mathcal{B}^{ST}$

---

accompanying code. We utilized either their default parameters or the best-reported parameters from their reported publications.

All experiments were conducted with the same early stopping mechanism to prevent overfitting. To ensure robust results, we repeated each experiment with 3 different random seeds and reported the average performance. All experiments were performed using NVIDIA A100 80G GPUs to ensure consistency in computational resources and reproducibility. For evaluations, we employed three standard evaluation metrics to comprehensively assess model performance: Mean Absolute Error (MAE), Root Mean Square Error (RMSE), and Mean Absolute Percentage Error (MAPE). Lower values across these metrics indicate better performance.

### C.2 Backbone and Baseline Details

**Backbone Architectures.** We conducted experiments using four different spatio-temporal graph neural network architectures as backbones to evaluate the robustness and versatility of our approach:

- **STGNN** [69]: Spatial-Temporal Graph Neural Network integrates graph convolution with 1D convolution to capture spatial dependencies and temporal dynamics simultaneously. It employs a

**Algorithm 2** Retrieval-Augmented Spatio-Temporal Prediction

---

**Require:** Input features $X$, adjacency matrix $A$, pattern libraries $\mathcal{B}^S, \mathcal{B}^T, \mathcal{B}^{ST}$, backbone $f_\Theta$, fusion weight $\gamma$, retrieval count $k$

**Ensure:** Enhanced prediction $\hat{Y}$

1: **Pattern Retrieval:**
2: Project query features $X' \leftarrow \text{PROJECTFEATURES}(X)$
3: **Multi-dimensional Retrieval:**
4: **for** each pattern type $t \in \{S, T, ST\}$ **do**
5:     Calculate similarity scores $s_j^t \leftarrow \exp(-\|X' - k_j^t\|^2)$ for all $k_j^t \in \mathcal{B}^t$
6:     Retrieve top-$k$ patterns $\mathcal{R}_t \leftarrow \{(k_j^t, v_j^t, s_j^t)| j \in \text{TopK}(s^t, k)\}$     ▷ Via Eq. (10)
7:     Compute weighted average $v_t^* \leftarrow \sum_{(k_j^t, v_j^t, s_j^t) \in \mathcal{R}_t} \text{Softmax}(s_j^t) \cdot v_j^t$
8: **end for**
9: **Knowledge Fusion:**
10: Current representation $Z_1 \leftarrow \text{MLP}_1(f_\Theta(X, A))$     ▷ Via Eq. (11)
11: Retrieved knowledge $Z_2 \leftarrow \text{MLP}_2(v_S^* \oplus v_T^* \oplus v_{ST}^*)$     ▷ Via Eq. (11)
12: Fused representation $Z \leftarrow \gamma \cdot Z_1 + (1 - \gamma) \cdot Z_2$     ▷ Via Eq. (12)
13: **Prediction:**
14: Final prediction $\hat{Y} \leftarrow \text{Decoder}(Z)$
15: **return** Enhanced prediction $\hat{Y}$

---

Table 5: Comparison of the overall performance of different methods (ASTGNN backbone).

| Category | Method | Metric | Air-Stream 3 | 6 | 12 | Avg. | PEMS-Stream 3 | 6 | 12 | Avg. | Energy-Stream 3 | 6 | 12 | Avg. |
|---|---|---|---|---|---|---|---|---|---|---|---|---|---|---|
| **Back-bone** | Pretrain | MAE | 18.48±0.69 | 21.32±0.49 | 24.58±0.41 | 21.11±0.55 | 14.32±0.37 | 15.55±0.29 | 18.16±0.24 | 15.76±0.30 | 10.66±0.06 | 10.67±0.08 | 10.69±0.10 | 10.67±0.07 |
| | | RMSE | 29.35±0.69 | 34.06±0.31 | 39.12±0.21 | 33.58±0.45 | 22.43±0.23 | 24.80±0.16 | 29.48±0.03 | 25.12±0.14 | 10.86±0.16 | 10.91±0.19 | 11.04±0.26 | 10.93±0.21 |
| | | MAPE (%) | 26.02±2.94 | 29.85±2.14 | 34.32±1.21 | 29.63±2.19 | 32.06±6.21 | 33.36±5.42 | 37.08±4.89 | 33.81±5.52 | 170.60±6.09 | 171.99±7.13 | 172.01±5.81 | 171.16±5.95 |
| | Retrain | MAE | 19.03±0.38 | 21.83±0.31 | 25.03±0.34 | 21.62±0.32 | 13.13±0.07 | 14.37±0.06 | 16.94±0.12 | 14.57±0.08 | 5.45±0.10 | 5.38±0.12 | 5.43±0.12 | 5.43±0.09 |
| | | RMSE | 29.88±0.66 | 34.55±0.46 | 39.54±0.34 | 34.05±0.50 | 21.23±0.12 | 23.49±0.10 | 27.86±0.19 | 23.77±0.13 | 5.63±0.07 | 5.62±0.08 | 5.79±0.08 | 5.69±0.04 |
| | | MAPE (%) | 25.59±0.68 | 29.35±0.58 | 33.80±0.50 | 29.14±0.55 | 18.78±1.18 | 20.45±1.15 | 24.53±1.17 | 20.88±1.13 | 57.86±7.56 | 58.05±6.98 | 59.28±6.42 | 58.39±7.00 |
| **Architecture-Based** | TrafficStream | MAE | 18.71±0.94 | 21.69±0.69 | 25.00±0.43 | 21.43±0.72 | 12.98±0.03 | 14.29±0.08 | 16.98±0.20 | 14.50±0.09 | 5.47±0.01 | 5.47±0.01 | 5.50±0.05 | 5.48±0.02 |
| | | RMSE | 29.04±1.01 | 34.21±0.72 | 39.51±0.45 | 33.57±0.77 | 21.07±0.09 | 23.51±0.18 | 28.20±0.41 | 23.81±0.20 | 5.60±0.00 | 5.64±0.01 | 5.78±0.04 | 5.67±0.01 |
| | | MAPE (%) | 24.29±1.52 | 28.26±0.98 | 33.10±0.54 | 28.09±1.04 | 17.43±0.67 | 19.21±0.42 | 23.51±0.38 | 19.65±0.30 | 58.91±6.22 | 59.53±5.90 | 60.87±5.34 | 59.62±5.85 |
| | ST-LoRA | MAE | 18.60±0.50 | 21.46±0.38 | 24.73±0.31 | 21.26±0.41 | 12.92±0.05 | 14.19±0.11 | 16.69±0.27 | 14.36±0.13 | 5.35±0.23 | 5.34±0.25 | 5.41±0.18 | 5.37±0.20 |
| | | RMSE | 29.25±0.70 | 34.16±0.41 | 39.33±0.25 | 33.62±0.50 | 20.89±0.16 | 23.17±0.28 | 27.42±0.64 | 23.41±0.33 | 5.50±0.20 | 5.53±0.21 | 5.72±0.13 | 5.59±0.16 |
| | | MAPE (%) | 23.69±0.88 | 27.71±0.67 | 32.58±0.46 | 27.55±0.69 | 18.06±0.50 | 19.92±0.47 | 24.48±0.91 | 20.40±0.43 | 49.91±1.86 | 50.77±1.25 | 52.62±1.63 | 50.98±1.73 |
| | STKEC | MAE | 19.02±0.33 | 21.86±0.31 | 25.09±0.27 | 21.66±0.31 | 13.29±0.12 | 14.48±0.06 | 17.15±0.06 | 14.73±0.09 | 5.32±0.13 | 5.32±0.12 | 5.35±0.12 | 5.32±0.12 |
| | | RMSE | 29.77±0.33 | 34.75±0.24 | 39.86±0.18 | 34.19±0.26 | 21.63±0.19 | 23.86±0.14 | 28.44±0.24 | 24.21±0.18 | 5.46±0.12 | 5.51±0.12 | 5.64±0.10 | 5.53±0.11 |
| | | MAPE (%) | 24.06±0.60 | 27.73±0.48 | 32.25±0.43 | 27.62±0.52 | 17.61±0.65 | 19.02±0.60 | 22.32±0.69 | 19.34±0.61 | 48.18±3.06 | 48.91±3.11 | 50.69±3.21 | 49.12±2.96 |
| | EAC | MAE | 19.29±1.39 | 22.04±0.89 | 25.14±0.59 | 21.87±1.06 | 13.21±0.13 | 14.36±0.05 | 16.75±0.26 | 14.54±0.07 | 5.37±0.14 | 5.42±0.10 | 5.47±0.10 | 5.42±0.11 |
| | | RMSE | 30.02±2.10 | 34.67±1.17 | 39.57±0.66 | 34.22±1.49 | 21.18±0.11 | 23.23±0.12 | 27.16±0.48 | 23.46±0.16 | 5.56±0.05 | 5.67±0.00 | 5.84±0.05 | 5.68±0.01 |
| | | MAPE (%) | 25.39±2.27 | 29.06±1.65 | 33.39±1.04 | 28.86±1.70 | 21.96±1.39 | 22.80±0.82 | 25.96±1.01 | 23.27±0.86 | 52.93±1.72 | 54.20±2.07 | 55.47±1.36 | 53.97±1.62 |
| | ST-Adapter | MAE | 18.73±0.70 | 21.73±0.53 | 24.96±0.30 | 21.46±0.52 | 12.81±0.01 | 13.96±0.06 | 16.20±0.18 | 14.11±0.07 | 5.40±0.11 | 5.38±0.14 | 5.42±0.09 | 5.41±0.10 |
| | | RMSE | 28.82±0.82 | 34.16±0.61 | 39.40±0.32 | 33.46±0.59 | 20.64±0.04 | 22.74±0.16 | 26.58±0.44 | 22.94±0.19 | 5.56±0.20 | 5.67±0.00 | 5.74±0.04 | 5.64±0.06 |
| | | MAPE (%) | 23.57±0.59 | 27.51±0.38 | 32.28±0.17 | 27.35±0.41 | 18.47±0.27 | 19.83±0.31 | 23.09±0.32 | 20.14±0.30 | 55.35±1.47 | 56.20±1.21 | 57.71±1.28 | 56.22±1.34 |
| | GraphPro | MAE | 18.53±0.73 | 21.56±0.58 | 24.89±0.47 | 21.31±0.62 | 12.88±0.06 | 14.13±0.12 | 16.58±0.24 | 14.29±0.13 | 5.39±0.23 | 5.35±0.27 | 5.38±0.19 | 5.39±0.21 |
| | | RMSE | 29.04±0.79 | 34.32±0.61 | 39.58±0.53 | 33.65±0.66 | 20.82±0.09 | 23.05±0.21 | 27.19±0.46 | 23.28±0.23 | 5.55±0.20 | 5.56±0.23 | 5.71±0.14 | 5.62±0.17 |
| | | MAPE (%) | **23.16±0.74** | 27.44±0.63 | 32.56±0.65 | 27.26±0.67 | 17.68±0.27 | 19.38±0.47 | 23.50±0.94 | 19.79±0.51 | 52.80±1.23 | 53.68±1.32 | 55.45±1.18 | 53.81±1.24 |
| | PECPM | MAE | 18.63±0.67 | 21.51±0.50 | 24.78±0.39 | 21.29±0.54 | 13.04±0.06 | 14.25±0.06 | 16.58±0.09 | 14.42±0.07 | 5.28±0.31 | 5.25±0.35 | 5.32±0.27 | 5.30±0.29 |
| | | RMSE | 29.26±0.83 | 34.22±0.52 | 39.39±0.34 | 33.65±0.61 | 21.06±0.09 | 23.24±0.12 | 27.31±0.16 | 23.47±0.12 | 5.43±0.27 | 5.46±0.30 | 5.64±0.20 | 5.52±0.23 |
| | | MAPE (%) | 23.75±1.00 | 27.75±0.70 | 32.60±0.47 | 27.58±0.75 | 17.95±0.14 | 19.57±0.19 | 23.44±0.56 | 19.96±0.19 | 52.31±4.48 | 53.28±4.30 | 54.94±4.32 | 53.34±4.34 |
| **Regularization-based** | EWC | MAE | 19.09±0.52 | 22.01±0.36 | 25.28±0.24 | 21.77±0.38 | 13.13±0.12 | 14.51±0.13 | 17.40±0.21 | 14.75±0.14 | 5.48±0.15 | 5.46±0.19 | 5.53±0.11 | 5.50±0.13 |
| | | RMSE | 29.62±0.81 | 34.75±0.49 | 40.06±0.22 | 34.14±0.55 | 21.41±0.32 | 24.00±0.39 | 29.15±0.59 | 24.37±0.42 | 5.63±0.13 | 5.67±0.15 | 5.85±0.06 | 5.72±0.09 |
| | | MAPE (%) | 24.51±0.85 | 28.33±0.67 | 32.99±0.54 | 28.16±0.69 | 17.35±0.62 | 19.29±0.59 | 23.83±0.70 | 19.75±0.59 | 52.49±0.98 | 53.32±1.00 | 54.91±0.85 | 53.45±0.90 |
| **Replay-based** | Replay | MAE | 18.53±0.51 | 21.51±0.38 | 24.83±0.22 | 21.26±0.39 | 13.02±0.02 | 14.34±0.03 | 17.05±0.09 | 14.55±0.04 | 5.23±0.15 | 5.22±0.18 | 5.28±0.13 | 5.25±0.14 |
| | | RMSE | 28.85±0.55 | 34.01±0.38 | 39.27±0.20 | 33.37±0.38 | 21.19±0.03 | 23.62±0.08 | 28.16±0.20 | 23.95±0.09 | 5.39±0.12 | 5.43±0.14 | 5.62±0.08 | 5.51±0.09 |
| | | MAPE (%) | 24.06±0.47 | 27.95±0.20 | 32.70±0.02 | 27.78±0.25 | 17.75±0.85 | 19.38±0.78 | 23.42±0.65 | 19.82±0.76 | 50.77±2.47 | 51.56±2.66 | 53.22±2.56 | 51.73±2.52 |
| **Retrieval-based** | STRAP | MAE | **18.05±0.70** | **20.61±0.57** | **23.54±0.39** | **20.44±0.57** | **12.26±0.03** | **13.31±0.05** | **15.53±0.11** | **13.49±0.05** | **4.89±0.04** | **4.90±0.04** | **4.92±0.03** | **4.90±0.03** |
| | | RMSE | **26.28±0.51** | **30.74±0.43** | **35.45±0.21** | **30.28±0.40** | **18.71±0.06** | **20.48±0.09** | **24.15±0.20** | **20.77±0.11** | **5.00±0.05** | **5.05±0.04** | **5.19±0.04** | **5.07±0.04** |
| | | MAPE (%) | 23.93±1.44 | **27.22±1.02** | **31.32±0.60** | **27.13±1.06** | **16.36±0.18** | **17.65±0.15** | **20.75±0.32** | **17.97±0.18** | **43.09±1.38** | **43.76±1.35** | **45.12±1.37** | **43.89±1.34** |

hierarchical structure where graph convolution extracts spatial features while temporal convolution captures sequential patterns, making it effective for traffic forecasting tasks.

- **ASTGNN** [18]: Attention-based Spatial-Temporal Graph Neural Network enhances STGNN with multi-head attention mechanisms to dynamically capture both spatial and temporal dependencies. It introduces adaptive adjacency matrices that evolve over time, allowing the model to identify important connections that might change across different time periods.

- **DCRNN** [37]: Diffusion Convolutional Recurrent Neural Network combines diffusion convolution with recurrent neural networks to model spatio-temporal dependencies. It formulates the traffic flow as a diffusion process on a directed graph and employs an encoder-decoder architecture with scheduled sampling for sequence prediction tasks.

Table 6: Comparison of the overall performance of different methods (DCRNN backbone).

| Category | Method | Metric | Air-Stream 3 | 6 | 12 | Avg. | PEMS-Stream 3 | 6 | 12 | Avg. | Energy-Stream 3 | 6 | 12 | Avg. |
|---|---|---|---|---|---|---|---|---|---|---|---|---|---|---|
| Backbone | Pretrain-ST | MAE | 21.52±2.34 | 23.67±2.05 | 26.30±1.72 | 23.61±2.07 | 16.41±0.34 | 16.75±0.25 | 18.60±0.16 | 17.09±0.25 | 10.58±0.02 | 10.61±0.02 | 10.68±0.11 | 10.63±0.05 |
| | | RMSE | 33.09±3.79 | 36.97±3.10 | 41.35±2.46 | 36.71±3.21 | 26.04±0.59 | 26.71±0.50 | 29.74±0.36 | 27.21±0.50 | 10.82±0.13 | 10.88±0.18 | 11.01±0.26 | 10.91±0.21 |
| | | MAPE (%) | 27.13±3.27 | 30.22±2.70 | 34.35±1.87 | 30.25±2.66 | 35.77±1.38 | 36.24±1.35 | 39.30±1.16 | 36.87±1.24 | 176.11±9.69 | 177.89±11.24 | 180.44±13.15 | 178.33±11.75 |
| | Retrain-ST | MAE | 22.14±1.45 | 24.26±1.22 | 26.79±0.98 | 24.17±1.24 | 14.16±0.17 | 14.73±0.10 | 16.81±0.11 | 15.05±0.13 | 5.26±0.02 | 5.21±0.09 | 5.19±0.14 | 5.21±0.10 |
| | | RMSE | 33.67±2.79 | 37.58±2.19 | 41.86±1.62 | 37.27±2.28 | 22.91±0.33 | 23.98±0.20 | 27.50±0.18 | 24.48±0.24 | 5.46±0.06 | 5.46±0.04 | 5.53±0.09 | 5.47±0.05 |
| | | MAPE (%) | 28.94±3.30 | 31.61±3.03 | 35.30±2.72 | 31.65±3.01 | 23.84±1.27 | 24.55±1.23 | 27.36±1.28 | 25.01±1.25 | 50.38±1.10 | 50.95±1.00 | 52.03±0.82 | 51.01±0.95 |
| Architecture-Based | TrafficStream | MAE | 18.40±0.64 | 21.26±0.53 | 24.54±0.43 | 21.08±0.54 | 13.64±0.08 | 14.56±0.08 | 16.70±0.08 | 14.78±0.08 | 5.35±0.07 | 5.29±0.13 | 5.28±0.16 | 5.29±0.13 |
| | | RMSE | 28.74±0.77 | 33.79±0.62 | 39.07±0.51 | 33.28±0.65 | 22.12±0.14 | 23.78±0.14 | 27.39±0.15 | 24.10±0.14 | 5.51±0.08 | 5.52±0.09 | 5.52±0.08 | 5.52±0.08 |
| | | MAPE (%) | 23.09±0.80 | 26.96±0.62 | 31.89±0.45 | 26.90±0.62 | 20.63±0.79 | 21.70±0.67 | 24.59±0.67 | 22.05±0.70 | 50.97±1.04 | 51.74±1.25 | 53.07±1.14 | 51.79±1.14 |
| | ST-LoRA | MAE | 19.65±0.45 | 22.17±0.43 | 25.16±0.39 | 22.05±0.42 | 13.07±0.10 | 13.96±0.06 | 16.02±0.06 | 14.16±0.08 | 5.28±0.07 | 5.23±0.10 | 5.21±0.12 | 5.23±0.10 |
| | | RMSE | 30.05±1.11 | 34.73±0.94 | 39.74±0.78 | 34.29±0.96 | 21.06±0.16 | 22.66±0.10 | 26.17±0.10 | 22.97±0.12 | 5.46±0.11 | 5.45±0.07 | 5.53±0.07 | 5.46±0.07 |
| | | MAPE (%) | 24.60±0.55 | 28.01±0.54 | 32.54±0.50 | 28.03±0.52 | 20.09±0.96 | 21.12±1.00 | 23.88±1.30 | 21.43±1.06 | 50.77±1.85 | 51.48±1.94 | 52.86±1.91 | 51.55±1.91 |
| | STKEC | MAE | 19.15±0.72 | 21.84±0.62 | 24.96±0.58 | 21.70±0.66 | 14.03±0.66 | 14.87±0.39 | 16.99±0.32 | 15.11±0.47 | 5.38±0.09 | 5.38±0.08 | 5.39±0.08 | 5.38±0.09 |
| | | RMSE | 29.65±1.05 | 34.45±0.79 | 39.46±0.66 | 33.97±0.88 | 23.29±0.97 | 24.76±0.55 | 28.31±0.44 | 25.13±0.66 | 5.51±0.08 | 5.55±0.08 | 5.66±0.08 | 5.56±0.09 |
| | | MAPE (%) | 23.85±0.87 | 27.58±0.68 | 32.41±0.49 | 27.58±0.66 | 19.82±1.44 | 20.73±1.18 | 23.44±1.22 | 21.08±1.28 | 52.87±1.08 | 53.45±1.08 | 54.70±1.09 | 53.57±1.07 |
| | EAC | MAE | 19.88±0.42 | 22.26±0.10 | 25.09±0.07 | 22.16±0.10 | 12.61±0.08 | 13.24±0.09 | 14.47±0.13 | 13.32±0.09 | 5.26±0.02 | 5.22±0.09 | 5.25±0.07 | 5.24±0.10 |
| | | RMSE | 30.23±0.13 | 34.63±0.18 | 39.39±0.15 | 34.26±0.15 | 20.11±0.14 | 21.22±0.14 | 23.25±0.20 | 21.33±0.16 | 5.48±0.08 | 5.48±0.08 | 5.60±0.06 | 5.51±0.07 |
| | | MAPE (%) | 24.66±0.43 | 27.81±0.41 | 32.04±0.33 | 27.85±0.39 | 18.61±0.50 | 19.34±0.50 | 20.90±0.59 | 19.47±0.51 | 54.10±3.71 | 55.23±2.95 | 55.23±2.97 | 55.23±2.97 |
| | ST-Adapter | MAE | 20.12±0.24 | 22.61±0.19 | 25.52±0.20 | 22.47±0.21 | 13.12±0.18 | 13.99±0.13 | 15.99±0.15 | 14.19±0.15 | 5.36±0.07 | 5.32±0.15 | 5.30±0.20 | 5.32±0.15 |
| | | RMSE | 30.56±0.51 | 35.28±0.46 | 40.20±0.38 | 34.80±0.46 | 21.04±0.30 | 22.63±0.22 | 26.07±0.27 | 22.92±0.26 | 5.57±0.05 | 5.58±0.11 | 5.65±0.16 | 5.58±0.11 |
| | | MAPE (%) | 24.93±0.72 | 28.15±0.42 | 32.53±0.24 | 28.20±0.46 | 20.31±0.49 | 21.23±0.34 | 23.65±0.24 | 21.50±0.36 | 52.95±2.44 | 53.99±2.61 | 55.13±2.71 | 53.91±2.67 |
| | GraphPro | MAE | 19.80±0.23 | 22.29±0.19 | 25.23±0.15 | 22.17±0.20 | 13.11±0.10 | 14.01±0.07 | 16.08±0.08 | 14.21±0.08 | 5.33±0.14 | 5.28±0.15 | 5.28±0.19 | 5.28±0.16 |
| | | RMSE | 30.03±0.58 | 34.76±0.47 | 39.75±0.36 | 34.30±0.49 | 21.11±0.17 | 22.70±0.09 | 26.20±0.11 | 23.01±0.11 | 5.52±0.15 | 5.52±0.14 | 5.61±0.16 | 5.53±0.14 |
| | | MAPE (%) | 24.43±0.39 | 27.79±0.32 | 32.32±0.29 | 27.83±0.33 | 20.13±0.62 | 21.25±0.63 | 24.00±0.82 | 21.54±0.65 | 50.23±2.79 | 50.23±2.79 | 51.71±2.63 | 50.33±2.77 |
| | PECPM | MAE | 19.47±0.74 | 22.06±0.60 | 25.11±0.47 | 21.93±0.62 | 13.32±0.16 | 14.19±0.13 | 16.19±0.12 | 14.38±0.14 | 5.30±0.06 | 5.24±0.03 | 5.21±0.05 | 5.23±0.04 |
| | | RMSE | 29.86±1.32 | 34.60±0.99 | 39.66±0.71 | 34.16±1.05 | 21.61±0.25 | 23.14±0.20 | 26.50±0.22 | 23.44±0.23 | 5.52±0.13 | 5.50±0.09 | 5.57±0.06 | 5.51±0.08 |
| | | MAPE (%) | 24.34±0.82 | 27.82±0.66 | 32.38±0.47 | 27.81±0.66 | 20.67±0.22 | 21.66±0.30 | 24.43±0.58 | 22.00±0.35 | 49.46±0.47 | 50.12±0.59 | 51.37±0.68 | 50.19±0.57 |
| Regularization-based | EWC | MAE | 18.61±0.91 | 21.43±0.76 | 24.67±0.62 | 21.26±0.78 | 14.35±0.15 | 15.24±0.14 | 17.37±0.11 | 15.46±0.14 | 5.22±0.06 | 5.16±0.07 | 5.12±0.11 | 5.16±0.08 |
| | | RMSE | 29.27±1.14 | 34.22±0.91 | 39.42±0.73 | 33.73±0.96 | 23.87±0.16 | 25.42±0.14 | 28.98±0.06 | 25.76±0.13 | 5.40±0.11 | 5.38±0.04 | 5.44±0.03 | 5.39±0.04 |
| | | MAPE (%) | 23.05±1.31 | 26.93±1.05 | 31.89±0.84 | 26.87±1.09 | 20.90±0.99 | 21.98±1.04 | 24.70±1.14 | 22.28±1.05 | 49.78±1.43 | 50.41±1.30 | 51.50±1.16 | 50.43±1.23 |
| Replay-based | Replay | MAE | 18.35±0.53 | 21.20±0.38 | 24.47±0.25 | 21.03±0.40 | 13.99±0.10 | 14.92±0.10 | 17.12±0.10 | 15.15±0.10 | 5.28±0.07 | 5.21±0.15 | 5.15±0.20 | 5.20±0.16 |
| | | RMSE | 28.73±0.61 | 33.71±0.46 | 38.96±0.37 | 33.23±0.49 | 22.81±0.19 | 24.41±0.15 | 28.09±0.12 | 24.78±0.15 | 5.45±0.04 | 5.42±0.09 | 5.46±0.14 | 5.43±0.10 |
| | | MAPE (%) | 23.09±0.42 | 26.90±0.19 | 31.82±0.13 | 26.85±0.22 | 22.16±0.25 | 23.22±0.30 | 26.06±0.34 | 23.56±0.29 | 50.14±2.63 | 50.49±2.50 | 51.59±2.43 | 50.61±2.49 |
| Retrieval-based | STRAP | MAE | 18.86±1.16 | 21.10±0.93 | 23.77±0.68 | 20.99±0.95 | 12.36±0.28 | 13.18±0.18 | 15.12±0.16 | 13.38±0.21 | 4.89±0.04 | 4.90±0.04 | 4.92±0.03 | 4.90±0.03 |
| | | RMSE | 27.24±1.61 | 31.35±1.31 | 35.82±0.99 | 31.00±1.34 | 18.76±0.43 | 20.17±0.27 | 23.42±0.25 | 20.48±0.33 | 5.00±0.05 | 5.05±0.04 | 5.19±0.04 | 5.07±0.04 |
| | | MAPE (%) | 23.79±0.93 | 26.75±0.56 | 30.74±0.22 | 26.77±0.62 | 18.13±0.46 | 18.82±0.36 | 21.07±0.40 | 19.14±0.38 | 43.09±1.38 | 43.76±1.35 | 45.12±1.37 | 43.89±1.34 |

Table 7: Hyperparameter Settings for All Methods Across Different Datasets

| Method | AIR-Stream LR | BS | PEMS-Stream LR | BS | ENERGY-Stream LR | BS |
|---|---|---|---|---|---|---|
| Pretrain | 0.03 | 128 | 0.03 | 128 | 0.01 | 128 |
| Retrain | 0.03 | 128 | 0.03 | 128 | 0.03 | 128 |
| TrafficStream | 0.01 | 128 | 0.03 | 128 | 0.03 | 128 |
| ST-LoRA | 0.03 | 128 | 0.03 | 128 | 0.03 | 128 |
| STKEC | 0.01 | 128 | 0.01 | 128 | 0.03 | 128 |
| EAC | 0.03 | 128 | 0.03 | 128 | 0.03 | 128 |
| ST-Adapter | 0.03 | 128 | 0.03 | 128 | 0.03 | 128 |
| GraphPro | 0.03 | 128 | 0.03 | 128 | 0.03 | 128 |
| PECPM | 0.03 | 128 | 0.03 | 128 | 0.03 | 128 |
| EWC | 0.01 | 128 | 0.03 | 128 | 0.03 | 128 |
| Replay | 0.01 | 128 | 0.03 | 128 | 0.03 | 128 |
| RAGraph | 0.03 | 128 | 0.03 | 128 | 0.03 | 128 |
| PRODIGY | 0.03 | 128 | 0.03 | 128 | 0.03 | 128 |
| **STRAP** | **0.03** | **128** | **0.03** | **128** | **0.03** | **128** |

- **TGCN** [80]: Temporal Graph Convolutional Network integrates graph convolutional networks with gated recurrent units to capture spatial dependencies and temporal dynamics simultaneously. It maintains a balance between model complexity and predictive power, making it particularly suitable for traffic prediction tasks with limited computational resources.

**Baseline Methods.** We compared our proposed STRAP framework against various state-of-the-art methods spanning multiple categories:

❶ **Backbone-based Methods.**

- **Pretrain**: This approach involves training the backbone model on historical data and directly applying it to new streaming data without any adaptation. It serves as a lower bound baseline that illustrates the performance degradation when models fail to adapt to distribution shifts.

- **Retrain**: This method completely retrains the backbone model from scratch whenever new data arrives. While it can adapt to new distributions, it suffers from catastrophic forgetting of previously learned patterns and incurs substantial computational costs.

❷ **Architecture-based Methods.**

- **TrafficStream** [10]: A streaming traffic flow forecasting framework based on continual learning principles. It maintains a memory buffer of historical samples and employs experience replay to mitigate catastrophic forgetting while adapting to evolving traffic patterns.

- **ST-LoRA** [54]: Spatio-Temporal Low-Rank Adaptation introduces parameter-efficient fine-tuning for spatio-temporal models by inserting lightweight, trainable low-rank matrices while keeping most pretrained parameters frozen, enabling efficient adaptation to new distributions with minimal computational overhead.

- **STKEC** [63]: Spatio-Temporal Knowledge Expansion and Consolidation framework specifically designed for continual traffic prediction with expanding graphs. It maintains knowledge from historical graphs while accommodating new nodes and edges through specialized knowledge transfer mechanisms.

- **EAC** [9]: Expand and Compress introduces a parameter-tuning framework for continual spatio-temporal graph forecasting that freezes the base model to preserve prior knowledge while adapting to new data through prompt parameter pools, effectively balancing stability and plasticity.

- **ST-Adapter** [49]: Originally designed for image-to-video transfer learning, this approach adapts pretrained models to spatio-temporal tasks by inserting lightweight adapter modules that capture temporal dynamics while preserving spatial representations from the pretrained model.

- **GraphPro** [71]: A graph pre-training and prompt learning framework that utilizes task-specific prompts to adapt pretrained graph neural networks to downstream tasks, achieving efficient knowledge transfer with minimal parameter updates.

❸ **Regularization-based Methods. EWC** [42]: Elastic Weight Consolidation selectively constrains important parameters for previously learned tasks while allowing flexibility in less critical parameters. It utilizes Fisher information matrices to estimate parameter importance and applies regularization accordingly, preventing catastrophic forgetting during continual learning.

❹ **Replay-based Methods. Replay** [16]: This approach maintains a memory buffer of representative samples from historical data distributions and periodically replays them during training on new data. It explicitly preserves knowledge of previous distributions, though at the cost of increased memory requirements and potentially inefficient knowledge consolidation.

❺ **Pattern Matching-based Methods. PECPM** [64]: Pattern Expansion and Consolidation on evolving graphs for continual traffic prediction leverages pattern matching techniques to identify relevant historical patterns. It constructs a pattern memory module that stores and retrieves patterns based on spatio-temporal similarity, enabling adaptation to evolving graph structures.

## C.3 Datasets Statistics

Our experiments use real-world natural streaming datasets, and detailed statistics for each dataset are shown in Table 8 below:

Table 8: Dataset Details

| Dataset | Domain | Time Range | Period | Node Evolution | Frequency | Frames |
|---------|--------|------------|--------|----------------|-----------|--------|
| Air-Stream | Weather | 01/01/2016 - 12/31/2019 | 4 | $1087 \rightarrow 1115 \rightarrow 1154 \rightarrow 1193 \rightarrow 1202$ | 1 hour | 34,065 |
| PEMS-Stream | Traffic | 07/10/2011 - 09/08/2017 | 7 | $655 \rightarrow 713 \rightarrow 786 \rightarrow 822 \rightarrow 834 \rightarrow 850 \rightarrow 871$ | 5 min | 61,992 |
| Energy-Stream | Energy | Unknown (245 days) | 4 | $103 \rightarrow 113 \rightarrow 122 \rightarrow 134$ | 10 min | 34,560 |

We use three real-world streaming graph datasets for our analysis. First, the *PEMS-Stream* dataset serves as a benchmark for traffic flow prediction, where the goal is to predict future traffic flow based on historical observations from a directed sensor graph. Second, we utilize the *AIR-Stream* dataset for meteorological domain analysis, which focuses on predicting future air quality index (AQI) flow based on observations from various environmental monitoring stations located in China. We segment the data into four periods, corresponding to four years. Third, the *ENERGY-Stream* dataset examines wind power in the energy domain, where the objective is to predict future indicators based on the generation metrics of a wind farm operated by a specific company (using temperature inside the turbine nacelle as a substitute for active power flow observations). This dataset is also divided into four periods, corresponding to four years, according to the appropriate sub-period dataset size.

For all datasets, we follow conventional practices [37] to define the graph topology. Specifically, we compute the adjacency matrix $A$ for each year using a thresholded Gaussian kernel, defined as follows:

$$A_{i[t]j} = \begin{cases} \exp\left(-\frac{d_{ij}^2}{\sigma^2}\right) & \text{if } \exp\left(-\frac{d_{ij}^2}{\sigma^2}\right) \geq r \text{ and } i \neq j \\ 0 & \text{otherwise} \end{cases} \tag{44}$$

where $d_{ij}$ represents the distance between sensors $i$ and $j$, $\sigma$ is the standard deviation of all distances, and $r$ is the threshold. Empirically, we select $r$ values of 0.5 and 0.99 for the air quality and wind power datasets, respectively.

## C.4 Hyper-parameter Study (RQ3)

**Fusion Ratio.** Figure 5 presents the effect of varying fusion ratios on different backbone architectures, revealing distinct optimal balancing points between historical knowledge and current observations. For ASTGNN, we observe a clear performance improvement as the fusion ratio increases to 0.9, with MAPE decreasing from 43.6% to 40.7% on average, suggesting this attention-based architecture benefits significantly from prioritizing current observations. In contrast, STGNN shows optimal performance at a moderate fusion ratio of 0.7, with performance degrading at both extremes.

DCRNN and TGCN display markedly different patterns. DCRNN performs best at fusion ratios of 0.3 and 0.7-0.9, indicating its dual-phase nature benefits from

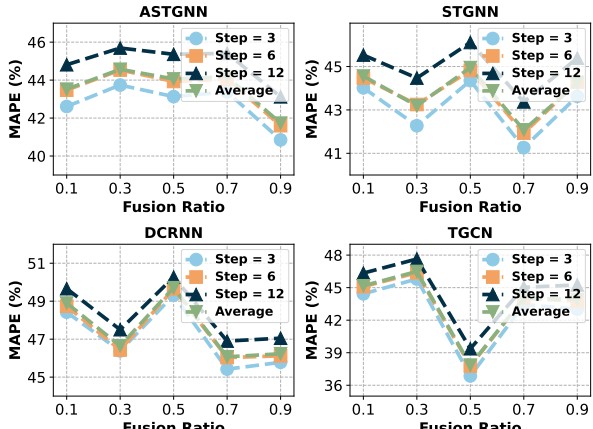

Figure 5: Performance analysis of different backbone across various fusion ratios $\gamma$.

either stronger historical knowledge incorporation or current observation emphasis, but struggles with equal weighting. Most notably, TGCN demonstrates the most pronounced sensitivity to the fusion ratio, achieving its optimal performance at precisely 0.5, with substantial performance degradation at other ratios. This suggests TGCN's temporal gating mechanisms particularly benefit from balanced integration of historical patterns and current data.

**Dropout Ratio.** Figure 6 demonstrates the impact of pattern library dropout rate on model performance across AST-GNN and DCRNN backbones. The consistently increasing MAPE with higher dropout rates provides compelling evidence for the effectiveness of our pattern library construction methodology. This clear degradation trend confirms that each pattern stored in our library contributes valuable information for accurate forecasting, with minimal redundancy or noise. The near-linear performance decline with increasing dropout suggests

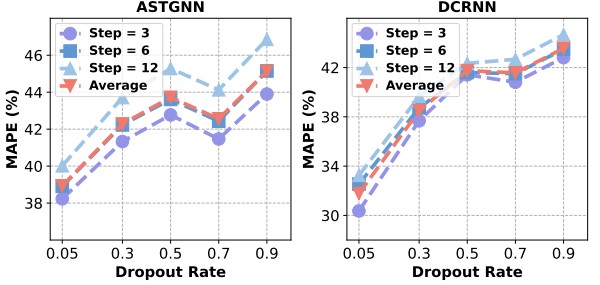

Figure 6: Performance analysis (MAPE) of different pattern library dropout ratios.

our pattern selection and extraction mechanisms effectively capture essential spatio-temporal dynamics while filtering out irrelevant information. The particularly steep performance drop in DCRNN further emphasizes the high information density of our constructed library, where even moderate pattern removal significantly impacts predictive capability.

**Retrieval Count.** Figure 7 examines how the number of retrieved patterns ($k$) from our pattern library affects prediction performance across STGNN and DCRNN backbones, revealing an optimal retrieval range that balances sufficient historical knowledge representation with minimal noise introduction. The results demonstrate a clear U-shaped relationship between retrieval count and prediction error, where both insufficient retrieval ($k = 5$) and excessive retrieval ($k = 70$) lead to suboptimal performance,

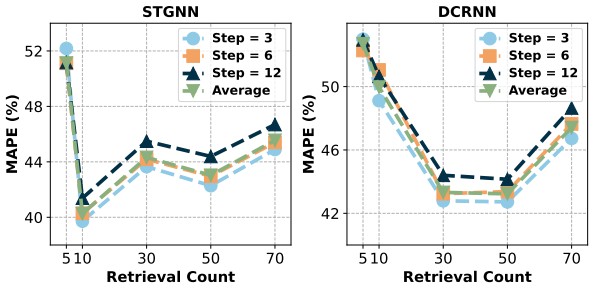

Figure 7: Performance analysis of different retrieval counts.

while moderate retrieval counts (10 for STGNN and 30-50 for DCRNN) achieve the lowest MAPE. This pattern suggests that retrieving too few patterns fails to capture sufficient historical knowledge for accurate predictions, while excessive retrieval introduces noise and potentially irrelevant patterns that dilute the quality of predictions. Notably, the optimal retrieval count differs between architectures, with the more complex DCRNN benefiting from a larger pattern set (30-50) compared to STGNN (10).

## C.5 Case Study (RQ4)

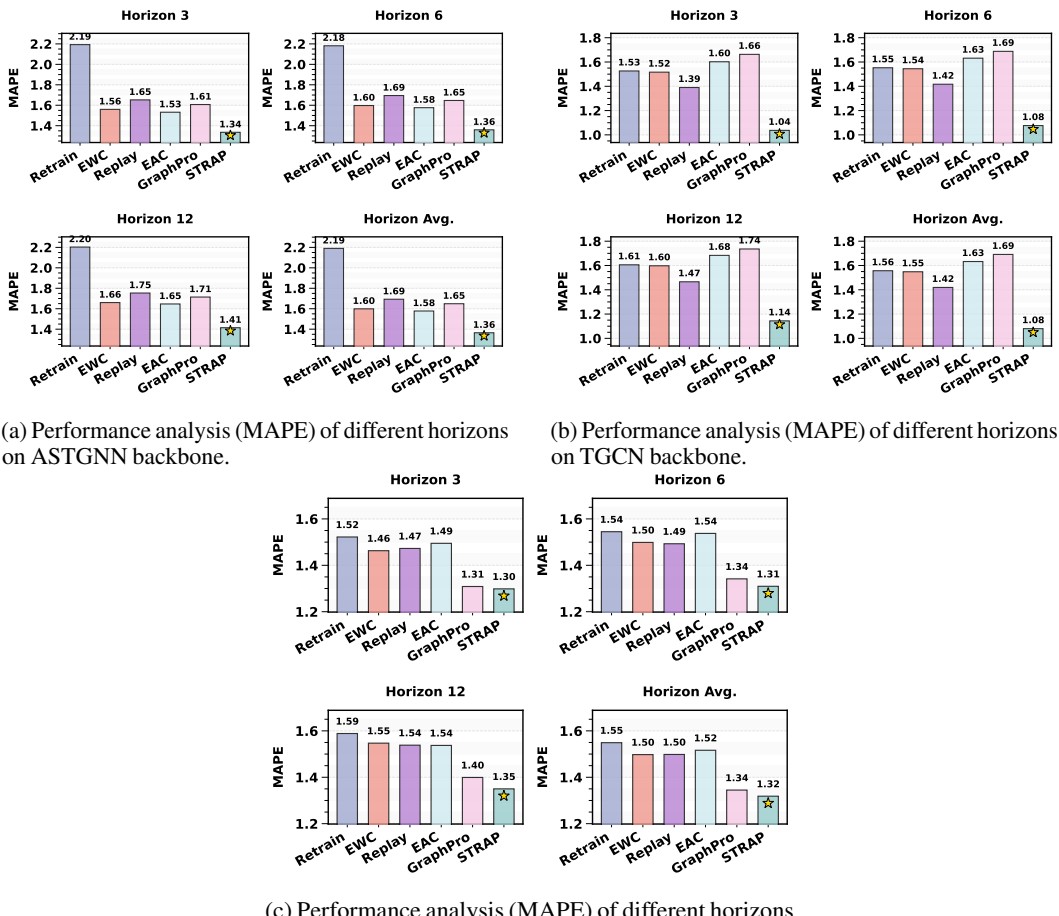

(a) Performance analysis (MAPE) of different horizons on ASTGNN backbone.

(b) Performance analysis (MAPE) of different horizons on TGCN backbone.

(c) Performance analysis (MAPE) of different horizons on DCRNN backbone.

Figure 8: Performance analysis (MAPE) of different horizons on different backbones.

As illustrated in Figure 8, we conducted a comprehensive analysis of model performance across different prediction horizons (3, 6, 12, and average) on three backbone architectures: ASTGNN, TGCN, and DCRNN. The results demonstrate that while baseline methods exhibit considerable performance variations across different backbone architectures, our STRAP consistently maintains superior performance regardless of the underlying backbone. This remarkable consistency can be attributed to two key factors: First, our approach decouples pattern extraction and retrieval from the specific neural network architecture, enabling a more robust knowledge representation that transcends the limitations of any particular backbone. Second, our multi-level pattern library framework operates as a plug-and-play enhancement that seamlessly integrates with various graph neural network foundations without requiring architecture-specific modifications. On ASTGNN, STRAP achieves average MAPE reductions of 19%, 22%, and 24% compared to the best baseline for horizons 3, 6, and 12, respectively. Similarly significant improvements are observed on TGCN and DCRNN. These consistent gains across different architectures underscore the versatility and robustness of our approach, which effectively enhances prediction performance without being constrained by backbone design choices or implementation details.

## C.6 Retrieval-based Methods and Computational Efficiency (RQ5)

### C.6.1 Retrieval-based Methods

We extended our experimental evaluation to incorporate two state-of-the-art methods, RAGraph [26] and PRODIGY [24], across all four backbone architectures (STGNN, ASTGNN, DCRNN, TGCN) on the ENERGY-Stream dataset (Figure 9). The comprehensive results demonstrate that while these advanced methods indeed provide substantial improvements over conventional baseline approaches, our proposed STRAP framework consistently achieves superior performance across all evaluation metrics and backbone architectures, further validating its effectiveness and generalizability.

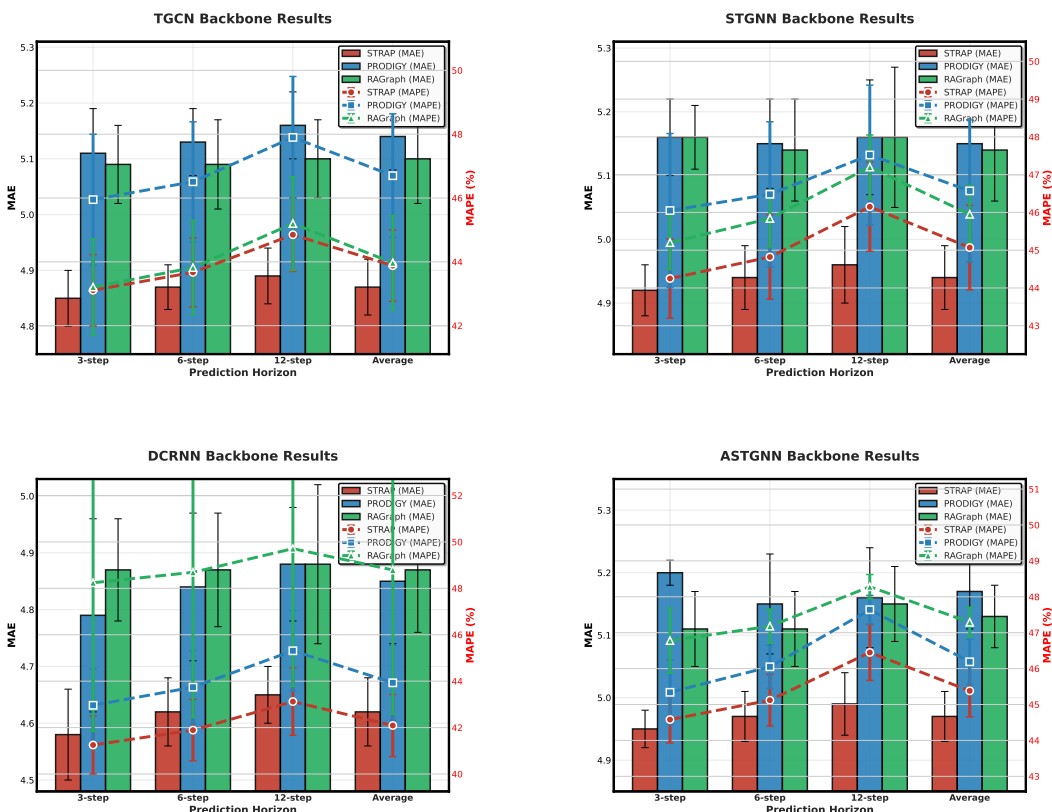

Figure 9: Performance comparison of STRAP, PRODIGY, and RAGraph across different backbone architectures. The figure shows MAE (bar charts) and MAPE (line charts) metrics for 3-step, 6-step, 12-step predictions and their averages. STRAP consistently outperforms both baseline methods across all backbone networks (TGCN, STGNN, DCRNN, ASTGNN).

### C.6.2 Computational Efficiency

As demonstrated in our experimental evaluation, STRAP achieves a favorable balance between computational efficiency and prediction performance. The computational cost analysis presented in Figure 10 reveals that STRAP's training time (measured in seconds per epoch) falls within a moderate range compared to existing methods. Specifically, using ASTGNN as the backbone architecture on the ENERGY-Stream dataset, our approach requires moderately higher computational resources than lightweight methods such as EAC, EWC, and GraphPro, but significantly less computational overhead than the most intensive baseline PECPM.

The inherent computational requirements of our retrieval-augmented framework stem from two primary components: pattern library maintenance and dynamic retrieval operations during training and inference. While this introduces unavoidable computational overhead compared to non-retrieval methods, the trade-off yields substantial benefits in prediction accuracy and enhanced robustness to distribution shifts—critical advantages in real-world traffic forecasting scenarios.

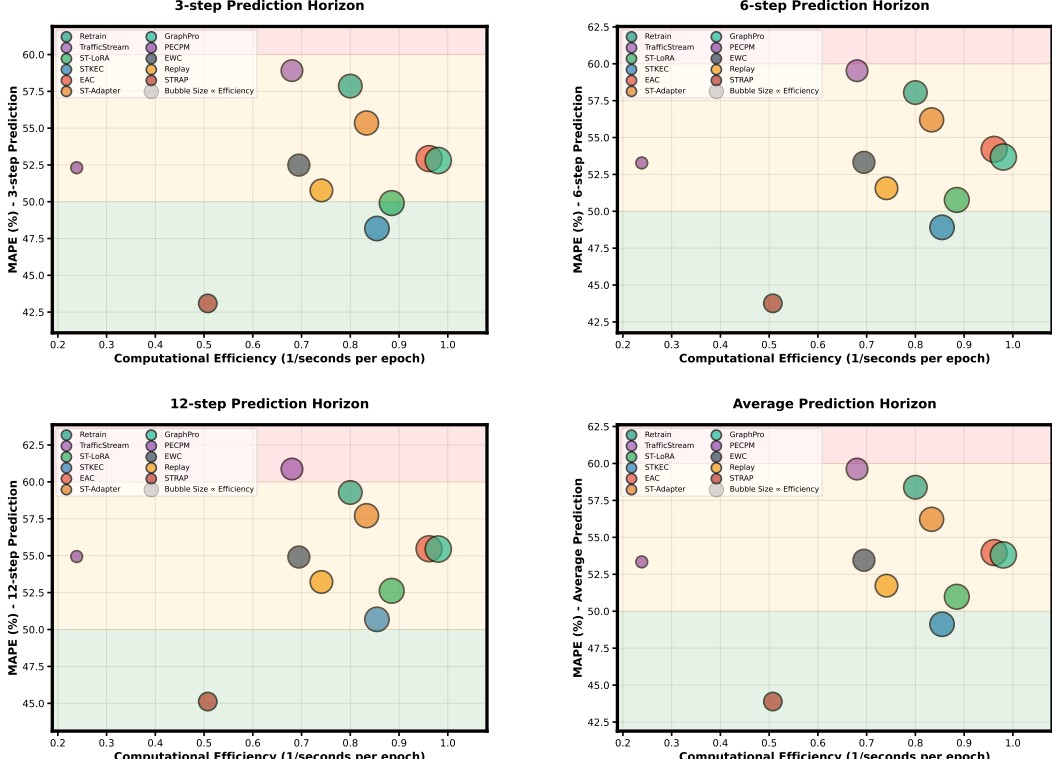

Figure 10: MAPE vs Computational Efficiency Analysis across Different Prediction Horizons. The bubble chart shows the relationship between computational efficiency (x-axis) and MAPE performance (y-axis) for all methods across 3-step, 6-step, 12-step, and average predictions on the ENERGY-Stream dataset. Bubble size represents computational efficiency, with larger bubbles indicating higher efficiency. STRAP consistently achieves the best MAPE performance across all prediction horizons while maintaining reasonable computational efficiency.

Despite the moderate computational cost increase, we argue that the performance gains justify this trade-off, particularly in applications where prediction accuracy under evolving traffic patterns is paramount. The computational overhead remains reasonable and scales efficiently with dataset size. Future research directions include exploring efficiency improvements through optimized pattern library indexing strategies and adaptive retrieval mechanisms to further optimize the performance-efficiency balance.

## D  Broader Impacts

Our work builds on the widespread application of retrieval-augmented methods in machine learning and aims to extend this paradigm to spatio-temporal graph data in streaming environments. This approach allows models to maintain consistent performance across distribution shifts without catastrophic forgetting, avoiding potential performance degradation common in streaming learning settings. Our retrieval-based framework is particularly effective in domains with continuous data streams and evolving patterns, such as traffic flow prediction, air quality monitoring, renewable energy forecasting, and intelligent transportation systems. Additionally, our multi-level pattern library establishes an excellent paradigm by explicitly storing and leveraging historical spatio-temporal patterns, significantly enhancing the model's adaptability to changing conditions. By combining retrieval mechanisms with spatio-temporal graph neural networks, our work provides valuable insights and serves as a reference for future streaming graph learning models in dynamic real-world applications.

# E   Data Ethics Statement

To evaluate the efficacy of this work, we conducted experiments using only publicly available datasets, namely, PEMS-Stream, AIR-Stream, and ENERGY-Stream datasets that contain traffic sensor data, air quality monitoring data, and wind power generation metrics, respectively. All datasets were used in accordance with their usage terms and conditions. We further declare that no personally identifiable information was used in these datasets. The traffic flow and air quality data were collected from public monitoring stations, while the energy data uses temperature measurements from wind turbines operated by an anonymized company with all identifying information removed. No human or animal subject was involved in this research, and all results reported reflect aggregated statistical measures without any privacy implications.

