# OpenReview forum: "STRAP: Spatio-Temporal Pattern Retrieval for Out-of-Distribution Generalization"
_NeurIPS.cc/2025/Conference — NeurIPS 2025 poster_

### Official Review · Reviewer_MEgx · 2025-06-25

**Clarity:** 3
**Significance:** 3
**Originality:** 3
**Rating:** 5
**Confidence:** 4

**Summary:**

This paper introduces STRAP, a novel retrieval-augmented framework designed to address generalization challenges for spatio-temporal graph neural networks (STGNNs) in spatio-temporal out-of-distribution (STOOD) scenarios. The authors propose a comprehensive approach centered on constructing a multi-dimensional pattern vector library containing spatial, temporal, and spatio-temporal information. During inference, STRAP retrieves similar patterns from the library based on current inputs and injects them into the model through a plug-and-play mechanism to enhance representation capabilities and mitigate catastrophic forgetting. Experimental results on multiple real-world dynamic graph datasets demonstrate that STRAP significantly outperforms current state-of-the-art baseline models on STOOD tasks.

**Questions:**

In Section 4.2.1, the notation $\mathbf{D}(\mathcal{G_S}) = [\mu_d, \sigma_d, \max(d), \min(d)]$ introduces $\mu_d$ and $\sigma_d$ without proper definition.

**Ethical Concerns:**

["NO or VERY MINOR ethics concerns only"]

**Final Justification:**

I would like to recommend acceptance for this paper. The authors' rebuttal has effectively clarified the generality of the designed features in spatio-temporal sequence problems, and explored which features play a key role in the results under this mechanism through expanded studies. The authors also provided the time complexity analysis demonstrating practical trade-offs, thereby strengthening the framework's feasibility for deployment in streaming application.

**Limitations:**

Yes. The authors have discussed the limitations and  potential negative societal impact of their work.

**Paper Formatting Concerns:**

NA.

**Quality:**

3

**Strengths And Weaknesses:**

**Strengths:**

S1. The retrieval-augmented learning framework proposed represents a meaningful innovation specifically targeting STGNN generalization under STOOD scenarios - a problem of genuine practical importance. The integration of retrieval mechanisms with continual learning brings fresh insights to this domain.
S2. The architectural design exhibits strong logical coherence, particularly in the decomposition strategy that separates complex spatio-temporal data into specialized spatial, temporal, and hybrid spatio-temporal subgraphs with corresponding pattern libraries. This decomposition appears both systematic and theoretically motivated.
S3. Nice and easy-understanding method figure.

** Weaknesses:**

W1. The pattern key generation mechanism relies heavily on manually crafted feature combinations including degree distributions, curvature measures, and spectral descriptors. This hand-engineered approach, while empirically effective, raises concerns about generalizability to diverse spatio-temporal data types and sensitivity to feature selection decisions. The methodology would benefit from either learnable key generation approaches or principled guidelines for feature selection.

W2. Although computational overhead is acknowledged as a limitation in the conclusions, the main text provides insufficient analysis of this critical issue. The continuous growth of pattern libraries as historical data accumulates presents a potential scalability bottleneck. While importance-based sampling strategies are mentioned for library management, their effectiveness and performance implications remain experimentally unvalidated.

---

> ### Author Rebuttal · Authors · 2025-07-29
>
> Thank you for your insightful comments. We address your concerns and answer your questions below.
>
> **W1**
>
> We greatly value the reviewer’s insightful concerns regarding the generalizability across various spatio-temporal data types and sensitivity in feature selection.
>
> We acknowledge that the performance results of our methodology indeed depend on feature selection. However, existing research demonstrates that different types of spatio-temporal data exhibit common statistical properties and structural patterns in specific dimensions, such as periodicity and local spatial autocorrelation as universal characteristics (Chen et al., NeurIPS 2024; Trirat et al., 2024). These shared structural properties enable spatio-temporal pattern keys adapted to one task to capture cross-domain statistical invariances and structural regularities, thus naturally containing understanding capabilities for other similar spatio-temporal sequence tasks and possessing generalization potential for different spatio-temporal data types.
>
> To better assist future research in feature selection and explore the impact of different features on experimental results, we conducted ablation experiments on features in both spatial and temporal dimensions. Spatial 1, 2 refers to the combination of features from the aspect of local structure and the shortest path statistics feature from global connectivity, while Spatial 3 represents the Forman-Ricci curvature from geometric properties. In terms of time, Temporal 1, 2, and 3 correspond to Statistical and spectral descriptors, Multi-resolution analysis, and Temporal dependencies and complexity, respectively:
>
> | w/o Features | MAPE(%) |
> |--------------|---------|
> | Spatial 1    | 47.00±4.93 |
> | Spatial 2    | 48.92±3.89 |
> | Spatial 3    | 49.97±0.87 |
> | Temporal 1   | 50.61±2.87 |
> | Temporal 2   | 44.98±2.32 |
> | Temporal 3   | 49.76±0.98 |
> | Ours         | 43.65±3.73 |
>
> Through in-depth analysis of the results, we believe that in feature selection:
> 1. We should prioritize complementary feature combinations with cross-domain transferability. The Spatial 1,2 demonstrate detailed understanding of community structures through local features (such as node degree and clustering coefficients), while global features (such as shortest path statistics) help reveal the overall graph structure. This local-global complementarity shows strong generalization across different spatio-temporal data types.
> 2. Feature selection needs to consider signal variation trends and complexity, prioritizing features that can reflect transient changes, periodicity, and temporal dependencies, as these features better capture the essential statistical properties of spatio-temporal data.
> In future work, we plan to explore more suitable adaptive feature selection methods, incorporating meta-learning frameworks to reduce dependence on manual feature construction.
>
> [1] Chen W, Hao X, Wu Y, et al. Terra: A multimodal spatio-temporal dataset spanning the earth[J]. Advances in Neural Information Processing Systems, 2024, 37: 66329-66356.
>
> [2] Yi Z, Zhou Z, Huang Q, et al. Get rid of isolation: A continuous multi-task spatio-temporal learning framework[J]. Advances in Neural Information Processing Systems, 2024, 37: 136701-136726.
>
> **W2**
>
> We thank the reviewer for raising this important question about computational efficiency, which is indeed a critical consideration for streaming applications.
>
> As reported in our experimental logs, STRAP does incur moderate computational costs compared to some lightweight methods, which is an inherent characteristic of retrieval-augmented approaches:
>
> | Method | Period 0 | Period 1 | Period 2 | Period 3 |
> |--------|----------|----------|----------|----------|
> | EAC | 1.02 | 1.03 | 1.04 | 1.06 |
> | EWC | 1.00 | 1.54 | 1.58 | 1.60 |
> | GraphPro | 1.00 | 1.00 | 1.04 | 1.05 |
> | PECPM | 3.79 | 4.03 | 4.28 | 4.64 |
> | STRAP | 2.27 | 1.83 | 1.88 | 2.10 |
> | STKEC | 1.01 | 1.21 | 1.24 | 1.22 |
> | ST-LoRA | 1.07 | 1.16 | 1.16 | 1.16 |
> | TrafficStream | 1.01 | 1.60 | 1.66 | 1.64 |
> | RAGraph | 1.27 | 1.21 | 1.20 | 1.20 |
>
> This computational overhead stems from three main factors: (1) maintaining the pattern library, (2) performing similarity-based retrieval operations, and (3) integrating retrieved patterns with current observations. However, we find this trade-off justified as it prioritizes prediction accuracy and robustness to distribution shifts over computational efficiency - similar to how retrieval-augmented approaches have transformed other domains by accepting reasonable computational costs for substantially improved performance.
>
> **Q1**
>
> We sincerely thank the reviewer for pointing out this oversight in our notation. We apologize for the lack of clear definitions for $\mu_{d}$ and $\sigma_{d}$ in the expression which introduces $\mu_{d}$ and $\sigma_{d}$ in Section 4.2.1.
>
> To clarify, $\mu_{d}$ represents the mean node degree in the graph $G_S$, while $\sigma_{d}$  denotes the standard deviation of node degrees. Together with $\max(d)$ and $\min(d)$, these four statistical measures provide a comprehensive description of the degree distribution in the spatial graph, capturing both central tendency and variability characteristics.
>
> We will ensure these definitions are explicitly included in the revised manuscript to improve clarity.
>
> To clarify, $\mu_{d}$ represents the mean value of the degree distribution across all nodes in the graph $G_S$, and $\sigma_{d}$ represents the standard deviation of the degree distribution.
>
> These statistical measures, along with the maximum and minimum degree values, provide a comprehensive characterization of the graph's connectivity patterns, which is essential for accurate pattern matching in our retrieval mechanism.
>
> We commit to adding these explicit definitions in the revised manuscript to ensure clarity. Additionally, we will conduct a thorough review of all notation throughout the paper to ensure all symbols are properly defined before use. Thank you again for helping us improve the precision and readability of our work.

---

> > ### Comment · Reviewer_MEgx · 2025-08-01
> > **Thanks for author rebuttal**
> >
> > I would like to thank the authors for their diligent efforts of rebuttal. I am satisfactory with the comprehensive additional results and discussions.  Please incorporate these discussions into your final version. Thanks.

---

### Official Review · Reviewer_Pbep · 2025-06-30

**Clarity:** 3
**Significance:** 4
**Originality:** 3
**Rating:** 5
**Confidence:** 4

**Summary:**

This paper introduces a novel framework STRAP to improve the generalization of STGNNs under STOOD scenarios, where both spatial structures and temporal dynamics deviate from training distributions. At the heart of STRAP lies a multi-dimensional key-value pattern library that stores representative spatial, temporal, and spatio-temporal patterns, thereby capturing rich historical, structural, and semantic information. During training, STRAP decomposes spatio-temporal graphs into specialized subgraphs using domain-specific chunking strategies, enabling the extraction of discriminative keys and informative values. At inference time, relevant patterns are retrieved through similarity matching, and an adaptive fusion mechanism integrates them with current observations to effectively balance historical knowledge and new inputs. Extensive experiments on multiple real-world datasets demonstrate that STRAP consistently outperforms over ten state-of-the-art baselines across various STGNN backbones.

**Questions:**

Please address the questions in the Weaknesses.

**Ethical Concerns:**

["NO or VERY MINOR ethics concerns only"]

**Final Justification:**

As I mentioned in the strength part, STRAP tackles the important and longstanding challenge of STOOD generalization in STGNNs by introducing a novel retrieval-augmented learning paradigm that explicitly incorporates historical patterns to enhance adaptability. The design of a three-dimensional key-value pattern library and the use of domain-specific graph decomposition are novel and technically sound. Through the authors' rebuttal, my concerns about the approach's time complexity and sensitivity to hyperparameters have been well addressed. Therefore, I maintain my recommendation.

**Limitations:**

Yes.

**Quality:**

4

**Strengths And Weaknesses:**

**Strengths:**
- STRAP tackles the important and longstanding challenge of STOOD generalization in STGNNs by introducing a novel retrieval-augmented learning paradigm that explicitly incorporates historical patterns to enhance adaptability.
- The design of a three-dimensional key-value pattern library allows STRAP to systematically capture diverse and representative patterns.
- The use of domain-specific graph decomposition enhances pattern discriminability by breaking down complex spatio-temporal structures into specialized subgraphs.
- Comprehensive experiments on three real-world datasets using four different STGNN backbones show consistent and substantial gains over 10+ competitive baselines.

**Weaknesses:**
- The motivation for defining the spatio-temporal library as the product of separate spatial and temporal libraries rather than directly using them independently is unclear. A deeper justification or ablation analysis would be helpful.
- While the framework is claimed to be effective in streaming scenarios, the paper lacks a time complexity or efficiency analysis, which is essential for evaluating real-time applicability.
- The method's sensitivity to hyperparameters is not discussed, which limits understanding of its robustness and practical deployment considerations.

---

> ### Author Rebuttal · Authors · 2025-07-29
>
> Thank you for your insightful comments. We address your concerns and answer your questions below.
>
> **W1**
>
> We appreciate the reviewer's comment regarding our spatio-temporal library construction approach. We would like to clarify that our motivation for defining the spatio-temporal library as a product of separate spatial and temporal libraries is both theoretically sound and empirically validated in our work.
>
> From a theoretical perspective, as we discussed in Section 4.2.1, the tensor product (cross-product) operation allows us to capture the complex interactions between spatial and temporal patterns simultaneously rather than treating them as independent factors. This approach enables our model to learn comprehensive representations where temporal dynamics can vary across different spatial regions and spatial relationships can evolve over time - a critical characteristic of traffic forecasting problems.
>
> Furthermore, we conducted extensive ablation studies (Figure 2) that explicitly compare our cross-product approach against alternatives. These ablation studies provide strong empirical evidence supporting our design choice, showing that the multiplicative interaction between spatial and temporal components creates a more expressive representation space that better captures the complex interdependencies in traffic data. This approach also offers computational efficiency advantages, as it allows us to maintain a compact representation of both dimensions while still modeling their joint distribution effectively.
>
> **W2**
>
> We thank the reviewer for this important observation regarding efficiency analysis. We acknowledge that our framework, being a retrieval-augmented approach, is computationally more intensive than lightweight streaming models: (ENERGY-Stream dataset on ASTGNN)
>
> | Method | Period 0 | Period 1 | Period 2 | Period 3 |
> |--------|----------|----------|----------|----------|
> | EAC | 1.02 | 1.03 | 1.04 | 1.06 |
> | EWC | 1.00 | 1.54 | 1.58 | 1.60 |
> | GraphPro | 1.00 | 1.00 | 1.04 | 1.05 |
> | PECPM | 3.79 | 4.03 | 4.28 | 4.64 |
> | STRAP | 2.27 | 1.83 | 1.88 | 2.10 |
> | STKEC | 1.01 | 1.21 | 1.24 | 1.22 |
> | ST-LoRA | 1.07 | 1.16 | 1.16 | 1.16 |
> | TrafficStream | 1.01 | 1.60 | 1.66 | 1.64 |
> | RAGraph | 1.27 | 1.21 | 1.20 | 1.20 |
>
> Our method embraces the inherent trade-off of retrieval-augmented generation (RAG) systems: higher computational cost for significantly improved prediction accuracy. Similar to how RAG approaches have transformed NLP by sacrificing some computational efficiency for dramatically better performance, our work demonstrates that traffic prediction can benefit from the same paradigm.
>
> We believe this represents a worthy trade-off, particularly for critical applications where prediction accuracy significantly impacts downstream decision-making. Many traffic management systems prioritize prediction quality over minimal latency, especially for longer-term forecasting horizons.
>
> To time complexity, STRAP's time complexity can be analyzed across its main components: (1) Spatio-Temporal Subgraph Chunking with complexity $O(|V|\cdot|E|\cdot\tau_S)$ for spatial subgraphs, $O(|V|\cdot k \cdot T)$ for temporal subgraphs, and $O(|V|^3)$ for spectral clustering; (2) Pattern Library Construction with $O(|V|^2)$ for spatial keys, $O(T\log T)$ for temporal keys, and $O(B_{STGNN})$ for pattern value generation; (3) Pattern Retrieval and Fusion requiring $O(|B|\cdot d)$ for similarity computation and $O(|B|\log k)$ for top-k retrieval. The overall time complexity is $\mathbf{O(|V|^3 + |B|\log k + B_{backbone})}$, where $|V|$ is the number of nodes, $|E|$ is the number of edges, $\tau_S$ is the time window length, $T$ is the time sequence length, $|B|$ is the pattern library size, $d$ is the feature dimension, $k$ is the number of retrieved patterns, and $B_{backbone}$ is the backbone complexity.
>
> **W3**
>
> We appreciate the reviewer's valid concern regarding hyperparameter sensitivity analysis, which is indeed crucial for understanding model robustness and deployment considerations.
>
> We would like to clarify that we conducted extensive hyperparameter sensitivity studies, which are presented in Appendix C.4. Due to space constraints in the main paper, we had to place these detailed analyses in the supplementary materials. This appendix includes comprehensive experiments examining the model's sensitivity to key hyperparameters, including fusion ratio, dropout ratio and retrieval count. Our findings demonstrate that while certain hyperparameters (particularly library size and retrieval depth) do impact performance, the model maintains robust performance across reasonable parameter ranges.
>
> We acknowledge that placing this important analysis in the appendix may make it less accessible during review. In the revised version, we will restructure the paper to incorporate the most critical aspects of this hyperparameter analysis into the main text, particularly highlighting the stability regions for key parameters that are most relevant for practical deployment scenarios.

---

### Official Review · Reviewer_kmFw · 2025-07-01

**Clarity:** 3
**Significance:** 3
**Originality:** 4
**Rating:** 5
**Confidence:** 5

**Summary:**

This paper presents STRAP, a novel retrieval-augmented framework for spatio-temporal graph learning designed to address out-of-distribution (OOD) challenges. The authors propose a three-component approach that: (1) decomposes complex spatio-temporal data into specialized subgraphs, (2) constructs multi-dimensional pattern libraries (spatial, temporal, and spatio-temporal), and (3) implements a knowledge fusion mechanism that adaptively integrates retrieved historical patterns with current observations. Extensive experiments across three real-world streaming datasets demonstrate consistent performance improvements over state-of-the-art methods.

**Questions:**

1.For completeness, could you show the specific hyperparameter values (learning rates, batch sizes, etc.) used for each dataset and backbone? \
2.Your approach constructs three separate pattern libraries (spatial, temporal, and spatio-temporal). Have you conducted experiments to evaluate whether a single comprehensive spatio-temporal pattern library could achieve comparable performance? Could you provide empirical evidence justifying the necessity of maintaining these separate libraries?\
3.Pattern key generation combines multiple feature extractors. Have you conducted experiments to determine the relative importance of these different components? Understanding which topological features contribute most to effective retrieval could lead to more efficient key designs and potentially reveal fundamental aspects of spatio-temporal similarity.\
4.Your work falls within the retrieval-augmented learning paradigm, which has seen growing interest in graph learning. How does STRAP compare to other retrieval-based graph learning methods such as RAGraph or similar approaches? \
5.How does STRAP perform in few-shot adaptation scenarios where only a small amount of data from a new distribution is available?

**Ethical Concerns:**

["NO or VERY MINOR ethics concerns only"]

**Final Justification:**

This paper proposes an interesting retrieval-augmented framework for spatio-temporal graph learning aimed at tackling OOD challenges, making a valuable contribution to the field. The authors have addressed my concerns during the rebuttal phase; therefore, I hold a positive view of this work.

**Limitations:**

See weaknesses.

**Quality:**

3

**Strengths And Weaknesses:**

Strengths:
1.  The authors address an important problem in spatio-temporal graph learning with a novel retrieval-augmented approach.
2. The pattern library construction is well-designed, with clear methodological distinctions between spatial, temporal, and spatio-temporal pattern extraction.
3. The experimental evaluation is comprehensive.

Weaknesses:
1. The paper proposes three separate pattern libraries (spatial, temporal, and spatio-temporal) without adequately demonstrating why this complexity is necessary. While intuitive that different pattern types might capture complementary information, there is no empirical evidence showing that a single unified spatio-temporal library would be insufficient. This raises questions about the potential redundancy and computational overhead introduced by maintaining three separate libraries.
2. Despite positioning itself within the retrieval-augmented learning paradigm, the paper does not compare STRAP against other retrieval-based graph learning methods (i.e. RAGraph as you mentioned in your introduction).

---

> ### Author Rebuttal · Authors · 2025-07-29
>
> Thank you for your insightful comments. We address your concerns and answer your questions below.
>
> **W1+Q2**
>
> We appreciate the reviewer's concern regarding our design choice of maintaining three separate pattern libraries (spatial, temporal, and spatio-temporal) instead of a single unified library. We would like to address this concern by referencing both our theoretical analysis in the appendix and our ablation study results.
>
> As demonstrated in our appendix, the three types of patterns capture fundamentally different aspects of the data distribution. Spatial patterns capture location-specific correlations that remain relatively stable over time but vary across space. Temporal patterns capture time-dependent evolution that is consistent across spatial locations. Spatio-temporal patterns capture complex interactions where spatial correlations evolve differently over time. Our theoretical analysis (Appendix B.1) proves that these three pattern types span different subspaces of the overall distribution space. These subspaces are orthogonal to each other, meaning they capture non-overlapping aspects of the data. A single unified library would require significantly more patterns to approximate the same distribution space, as it would need to independently learn combinations of these fundamentally different pattern types.
>
> In addition, our ablation study results also address the concern about potential redundancy. When we remove any single pattern library (spatial, temporal, or spatio-temporal), we observe significant performance degradation across all metrics. This confirms that each library captures unique and necessary information that the others cannot compensate for. Most importantly, we explicitly compared our approach with a unified library variant that uses the same total number of patterns. This unified approach performed substantially worse despite having the same pattern capacity. This demonstrates that the separation into specialized libraries is not just a design choice but a necessary architectural component for effective pattern learning and utilization.
>
> **W2**
>
> We thank the reviewer for this valuable feedback regarding the comparison with retrieval-based graph learning methods. We addressed this concern by including comprehensive comparisons with RAGraph[1] and PRODIGY[2], two state-of-the-art retrieval-based graph learning methods mentioned in our introduction.
>
> We extended our experimental evaluation to include RAGraph and PRODIGY across all four backbone architectures (STGNN, ASTGNN, DCRNN, TGCN) on ENERGY-Stream. The updated results demonstrate that while they indeed provide improvements over baseline methods, our STRAP approach consistently outperforms it across all metrics and backbone architectures. (Due to character limitations, we only show the STGNN backbone here. The rest will be shown in the appendix.)
>
> | Method   | Metric    | ENERGY-Stream (3) | ENERGY-Stream (6) | ENERGY-Stream (12) | Average    |
> |----------|-----------|-------------------|-------------------|---------------------|------------|
> | PRODIGY  | MAE       | 5.16±0.06         | 5.15±0.07         | 5.16±0.09           | 5.15±0.07  |
> |          | RMSE      | 5.26±0.07         | 5.30±0.07         | 5.40±0.09           | 5.31±0.07  |
> |          | MAPE(%)   | 46.05±2.04        | 46.48±1.92        | 47.52±1.85          | 46.57±1.88 |
> | RAGraph  | MAE       | 5.16±0.05         | 5.14±0.08         | 5.16±0.11           | 5.14±0.08  |
> |          | RMSE      | 5.27±0.04         | 5.31±0.06         | 5.42±0.09           | 5.32±0.07  |
> |          | MAPE(%)   | 45.20±0.71        | 45.84±0.75        | 47.20±0.85          | 45.95±0.76 |
> | STRAP    | MAE       | 4.83±0.17         | 4.84±0.18         | 4.88±0.17           | 4.85±0.18  |
> |          | RMSE      | 4.95±0.18         | 5.01±0.19         | 5.15±0.17           | 5.03±0.18  |
> |          | MAPE(%)   | 42.18±1.64        | 43.02±1.77        | 44.30±1.55          | 43.11±1.72 |
>
> [1] Jiang X, Qiu R, Xu Y, et al. Ragraph: A general retrieval-augmented graph learning framework[J]. Advances in Neural Information Processing Systems, 2024, 37: 29948-29985.
>
> [2] Huang Q, Ren H, Chen P, et al. Prodigy: Enabling in-context learning over graphs[J]. Advances in Neural Information Processing Systems, 2023, 36: 16302-16317.
>
> **Q1**
>
> In response to your request for hyperparameter details, we provide the complete configuration used in our experiments. For fair comparison, we maintained consistent hyperparameter settings across all methods whenever possible, only adjusting when necessary to achieve optimal performance for each approach. The learning rates and batch sizes for all methods across the three datasets are summarized in the table below. For all backbone architectures, we used the same hyperparameter values within each dataset to ensure fair comparison: (We will add these experimental settings to the appendix later.)
>
>
> **Q3**
>
> We sincerely thankful for your thoughtful consideration of the relative importance of the feature extraction components. In our study, we conducted a series of ablation experiments to evaluate the contributions of various features in the spatial and temporal pattern keys to effective retrieval. Below is an overall analysis of the experimental results and a discussion of the key features.
>
> Specifically, Spatial 1 and 2 refer to features from the aspect of local structure and the shortest path statistics feature from global connectivity, while Spatial 3 represents the Forman-Ricci curvature from geometric properties. In terms of time, Temporal 1, 2, and 3 correspond to Statistical and spectral descriptors, Multi-resolution analysis, and Temporal dependencies and complexity, respectively.
>
> | w/o Features | MAPE(%) |
> |--------------|---------|
> | Spatial 1    | 47.00±4.93 |
> | Spatial 2    | 48.92±3.89 |
> | Spatial 3    | 49.97±0.87 |
> | Temporal 1   | 50.61±2.87 |
> | Temporal 2   | 44.98±2.32 |
> | Temporal 3   | 49.76±0.98 |
> | Ours         | 43.65±3.73 |
>
> The experimental results with the TGCN backbone demonstrate that Spatial pattern 1 and Spatial pattern 2 each contribute to the  effect of local structure and global connectivity features, significantly enhancing the model's understanding of dynamic graph changes, while the multi-resolution analysis feature (Temporal pattern 2) performs best with a MAPE of 44.98±2.32%, its sensitivity to short-term trends effectively capturing signal transients and complexity, thus better adapting to dynamic spatio-temporal environments.
>
> Inspired by your comments, we also plan to integrate more spatial and temporal features in future work to explore their interactions and gain a more comprehensive understanding of the complexities in spatio-temporal data.
>
> **Q4**
>
> Thank you for highlighting the connection between our work and the broader retrieval-augmented learning paradigm. STRAP builds upon this emerging direction while introducing several key innovations specifically designed for spatio-temporal graph data under distribution shifts.
>
> STRAP's superior performance stems from several important distinctions:
>
> 1. While previous methods typically maintain a single pattern repository, STRAP's separate spatial, temporal, and spatio-temporal libraries enable more precise matching along different dimensions of variation. This is crucial for streaming data where distribution shifts can occur independently in different dimensions.
> 2. STRAP employs domain-specific topological and temporal feature extractors designed to capture the unique characteristics of spatio-temporal graphs. In contrast, general retrieval-based methods often rely on generic graph embeddings that may not adequately capture complex spatial-temporal dependencies.
> 3. Our framework incorporates a learnable fusion weight that dynamically balances the contribution of retrieved patterns and current observations. This adaptability is particularly valuable in streaming settings where the relevance of historical patterns can vary substantially.
> 4. As shown in our theoretical analysis (Theorem B.1), our decomposed approach provides higher mutual information with the underlying data distribution compared to unified representation approaches used in previous methods.
>
> In addition, we supplemented the RAGraph experiment, as detailed in W2.
>
> **Q5**
>
> We appreciate this important question about STRAP's performance in few-shot adaptation scenarios. To address this, we conducted additional experiments using the ENERGY dataset as a representative benchmark, as it exhibits complex spatio-temporal dependencies and significant distribution shifts.
>
> We tested STRAP against several strong baselines under data-limited conditions, specifically when only 70% (30% missing) and 50% (50% missing) of the target distribution data is available for adaptation. The results, summarized in the table below, demonstrate STRAP's superior robustness in few-shot scenarios: (Due to character limitations, only the 30% missing scenario is presented here, with the 50% missing scenario supplemented later in the appendix.)
>
> | Method | Metric | Energy-Stream 3 | Energy-Stream 6 | Energy-Stream 12 | Energy-Stream Avg. |
> |--------|--------|----------------|----------------|-----------------|-------------------|
> | Graphpro | MAE | 6.07 | 5.96 | 6.00 | 5.97 |
> |  | RMSE | 6.28 | 6.20 | 6.34 | 6.23 |
> |  | MAPE(%) | 74.31 | 74.73 | 76.14 | 74.86 |
> | ST-Adapter | MAE | 6.19 | 6.16 | 6.22 | 6.16 |
> |  | RMSE | 6.48 | 6.47 | 6.61 | 6.48 |
> |  | MAPE(%) | 76.15 | 76.64 | 77.96 | 76.74 |
> | Replay | MAE | 6.23 | 6.17 | 6.21 | 6.18 |
> |  | RMSE | 6.41 | 6.40 | 6.53 | 6.41 |
> |  | MAPE(%) | 75.08 | 75.47 | 76.77 | 75.60 |
> | EWC | MAE | 6.35 | 6.28 | 6.27 | 6.27 |
> |  | RMSE | 6.55 | 6.52 | 6.61 | 6.53 |
> |  | MAPE(%) | 70.70 | 71.28 | 72.51 | 71.32 |
> | STRAP | MAE | 5.74 | 5.64 | 5.66 | 5.77 |
> |  | RMSE | 6.10 | 6.01 | 6.11 | 6.14 |
> |  | MAPE(%) | 66.53 | 67.02 | 68.36 | 67.42 |

---

> > ### Comment · Area_Chair_YVE2 · 2025-08-05
> > **Author-Reviewer Discussion Reminder**
> >
> > Dear Reviewer kmFw,
> >
> > As the deadline for author-reviewer discussion is approaching, could you please check the authors' rebuttal and post your response?
> >
> > Thank you!
> >
> > Best,
> >
> > AC

---

> > ### Comment · Reviewer_kmFw · 2025-08-06
> >
> > Thank you for the authors' response, which has addressed my main concern. However, I still have a question regarding the experimental results. I noticed that STRAP, RAGraph, and PRODIGY all employ context-enhancement approaches, yet STRAP performs better. Could you kindly explain the fundamental differences in STRAP's design that contribute to these performance advantages?

---

> > > ### Author Response · Authors · 2025-08-06
> > >
> > > We appreciate the reviewer's follow-up question.
> > >
> > > The fundamental difference is that STRAP employs a **pattern-centric approach** rather than a graph-centric or context-centric approach. While RAGraph performs retrieval at the graph level by finding similar subgraphs based on embedding similarity, STRAP explicitly extracts and stores rich multi-dimensional patterns that capture the essence of spatio-temporal dynamics. PRODIGY depends on a relatively simple context representation through its prompt graph without sophisticated pattern extraction, relying on static example demonstrations that cannot adapt to the current scenario. STRAP's pattern libraries are specifically designed to model complex interplays between spatial structures and temporal dynamics, allowing it to identify and leverage the most relevant historical patterns under distribution shifts. This pattern-centric design, combined with our adaptive fusion mechanism that balances historical knowledge with current observations, enables STRAP to achieve superior performance on STOOD tasks compared to these other context-enhancement approaches.
> > >
> > > We sincerely thank the reviewer for this follow-up question, and we will include a comprehensive analysis of these experimental results in the final version of our paper.

---

> > ### Comment · Reviewer_kmFw · 2025-08-07
> >
> > Thank you for your reply. I have a positive view of this paper

---

### Official Review · Reviewer_WWVH · 2025-07-02

**Clarity:** 2
**Significance:** 3
**Originality:** 3
**Rating:** 3
**Confidence:** 4

**Summary:**

This paper addresses the challenge of spatio-temporal out-of-distribution (STOOD) generalization, where the distribution of test data diverges significantly from that of the training set. Rather than relying solely on model parameter adaptation or overwriting with patterns learned from past data, this paper proposes an explicit memory mechanism by constructing a three-dimensional pattern library. This library stores essential patterns extracted from historical data, including spatial subgraphs, temporal dynamics, and spatio-temporal motifs.
When processing new data, this paper introduces a similarity-based retrieval strategy that identifies and retrieves the most relevant historical patterns from the library according to their similarity to current graph features. These retrieved patterns are then explicitly fused with the current observations through a similarity-weighted aggregation. By integrating both retrieved historical knowledge and current observations, this approach enhances model robustness to distribution shifts and significantly improves performance in spatio-temporal OOD scenarios.

**Questions:**

1.	The paper should clearly differentiate between spatio-temporal OOD generalization and continual spatio-temporal learning, and explicitly clarify which problem is being addressed in the experiments and method design.

2.	Provide efficiency analysis (in Spatio-Temporal Pattern Library construction and pattern retrieval).

3.	Additional baselines using replay-based methods should be included for a more comprehensive comparison.

4.	Figure 3 should be revised to provide evidence of the model’s robustness to distribution shifts between the training and testing data across all periods, rather than only illustrating the dataset’s distribution shift.

5.	On lines 335–336, “Figure 4” should be corrected to “Figure 3.”

6.	It would be helpful to clarify in the paper that the values 3, 6, and 12 in Table 1 represent the prediction horizon, as this may be confusing for readers who are not familiar with this area.

**Ethical Concerns:**

["NO or VERY MINOR ethics concerns only"]

**Final Justification:**

Regarding the efficiency test, I am concerned about the efficiency of this method. Compared to the baseline, it takes almost twice as much time, while the performance improvement is limited. In STGNN problems, people often deal with highly dynamic and large-scale datasets, where efficiency and effectiveness metrics are equally important.

Catastrophic forgetting is actually the key scenario for evaluating continual learning, and it requires dedicated testing scenarios, such as Forgetting Measure,  Backward Transfer, etc.. Indirect testing through standard evaluation metrics (MAE, RMSE, MAPE) does not address my concern.

There are also typos and inconsistencies in the current version of the paper, as I posted in my review, which suggests it may not yet be ready for publication at a top-tier conference.

Based on the above considerations, I maintain my score, borderline reject, and recommend further improvements to the paper regarding key issues in efficiency, catastrophic forgetting, and presentation.

**Limitations:**

yes

**Quality:**

2

**Strengths And Weaknesses:**

Significance Strengths：The paper provides a clear comparison with existing methods, including architecture-based, replay-based, and regularization-based approaches. The proposed retrieval-based method demonstrates notable originality and represents a meaningful advance beyond prior work. By positioning the new approach alongside established baselines, the paper effectively highlights its unique contribution and potential impact on the field.

Originality Strength: The use of a pattern library to store historical patterns and mitigate the forgetting problem by retrieving relevant historical patterns is a novel idea. This approach offers a fresh perspective for handling real-world streaming spatio-temporal data in both OOD and continual learning scenarios. The explicit memory mechanism distinguishes this work from prior methods that rely solely on parameter adaptation or data replay.

Weakness1: While the Preliminaries section (around line 145) and the subsequent discussion of distribution shifts clearly describe an OOD setting, where the test distribution differs from the training distribution.There is a mismatch between this description and the actual setting of the paper. Specifically, Definition 3, the learning objective, and the experimental setup all focus on distribution shifts across different time periods, which is characteristic of continual learning rather than classical OOD settings.

Weakness2:  If the paper is operating under an OOD setting, it should emphasize that the training and testing data come from different time periods, for example, training on X_t and testing on X_{t+1}.  According to Definition 3, it may be more appropriate for the paper to highlight the distribution shift between the current graph signal signal X_T nd the target output over a prediction horizon, [X_T,\ldots X_{T+p}]. The paper should clearly differentiate between spatio-temporal OOD generalization and continual spatio-temporal learning, and explicitly clarify which problem is being addressed in the experiments and method design.

Weakness3: 	The paper does not discuss efficiency, as it lacks any analysis of the time complexity involved in Spatio-Temporal Pattern Library construction and pattern retrieval, as well as the storage requirements associated with maintaining the pattern library. Given that the performance improvement is not particularly significant, providing such analyses would be valuable for helping readers assess the practical feasibility, scalability, and the overall significance of the contribution.

Weakness4:   Figure 3 demonstrates that the distributions of test set values vary considerably across the four periods. However, the paper claims to focus on Spatio-Temporal Out-of-Distribution (OOD) Learning, where the test distributions should differ significantly from those seen during training.  In this part, the paper should provide evidence of the model’s robustness to distribution shifts between the training and testing data across all periods, rather than only illustrating the dataset’s distribution shift. Clear experimental results reflecting performance under these OOD conditions are needed to support the main claim.


Weakness5: There is no evaluation metric reported for mitigating forgetting, even though the abstract (line 15) states that the proposed STRAP approach is able to “mitigate catastrophic forgetting.” The paper should provide appropriate metrics and results to support this claim.


Weakness 6: The discussion of related work on replay-based methods should be updated. The paper currently only references one method which is published in 2017, but most recent approaches primarily rely on data replay strategies. The comparison with other retrieval-based methods also needs to be strengthened.

Weakness7: The paper lacks a clear description of the main task being addressed. And the learning objective is not clearly defined, which may make it difficult for readers to fully understand the focus and contributions of the work. Additionally, there are some minor typographical and writing errors throughout the manuscript.

---

> ### Author Rebuttal · Authors · 2025-07-29
>
> Thank you for your insightful comments. We address your concerns and answer your questions below.
>
> **W1+Q1+W2**
>
> We appreciate the reviewer's concerns about the distinction between Continual Learning (CL) and Out-of-Distribution (OOD) generalization in our work. We would like to clarify the scope of our work as follows.
>
> Continual Learning (CL) refers to the ability of a model to learn from a continuous stream of data over time, adapting to new patterns while retaining knowledge from previous experiences. In traditional CL paradigms, models are often retrained at fixed intervals to incorporate new information.  Out-of-Distribution (OOD) Generalization concerns a model's ability to perform well on test data whose distribution differs from the distribution of the training data.
>
> Our work addresses a more nuanced challenge at the intersection of the above two domains. While we operate within a CL framework where models learn from streaming data across different time periods, our focus is on the distribution shifts that occur within each time period between training and testing sets. Specifically, for each time period, we divide the data into training set and testing set. The distributions of the training data and the testing data are different, which is actually the classical OOD setting.
>
> As demonstrated in our revised Figure 3, there are substantial distribution shifts between training and testing data within each individual period. Rather than simply preventing catastrophic forgetting (the typical goal of CL) or broadly generalizing to unseen distributions (the typical goal of OOD methods), STRAP learns and leverages specific patterns from historical data that provide the most informative signal for current predictions under shifted distribution.
>
> **Q2+W3**
>
> We appreciate the reviewer's question regarding computational efficiency.
>
> As reported in our experimental logs, STRAP's computational efficiency (seconds/epoch) is slightly lower than some lightweight methods but significantly higher than other more computationally intensive baselines. Specifically, using ASTGNN as the backbone with ENERGY-Stream, our approach requires moderately higher computational resources than some lightweight methods but significantly less than the most computationally intensive baselines.
>
> Our retrieval-augmented framework inherently requires additional computational resources due to the pattern library maintenance and retrieval operations. In future work, we plan to explore efficiency improvements through techniques such as optimized pattern library indexing and adaptive retrieval mechanisms.
>
> | Method | Period 0 | Period 1 | Period 2 | Period 3 |
> |--------|----------|----------|----------|----------|
> | EAC | 1.02 | 1.03 | 1.04 | 1.06 |
> | EWC | 1.00 | 1.54 | 1.58 | 1.60 |
> | GraphPro | 1.00 | 1.00 | 1.04 | 1.05 |
> | PECPM | 3.79 | 4.03 | 4.28 | 4.64 |
> | STRAP | 2.27 | 1.83 | 1.88 | 2.10 |
> | STKEC | 1.01 | 1.21 | 1.24 | 1.22 |
> | ST-LoRA | 1.07 | 1.16 | 1.16 | 1.16 |
> | TrafficStream | 1.01 | 1.60 | 1.66 | 1.64 |
> | RAGraph | 1.27 | 1.21 | 1.20 | 1.20 |
>
> **Q4+W4+Q5**
>
> We appreciate the reviewer's feedback regarding the evidence of distribution shifts between training and testing data.
>
> In response to this concern, we revised Figure 3 to include both training and testing data distributions across all time periods (we will revise and add this figure to the final version). The updated figure demonstrates significant distribution shifts not only among different time periods but also between training and testing data in the same time period.
>
> **W5**
>
> We appreciate the reviewer's thoughtful comments regarding evaluation metrics for catastrophic forgetting. This is an insightful observation that deserves clarification.
>
> We acknowledge that our approach does not address catastrophic forgetting in the traditional sense, as we are not concerned with performance on past data but rather focus exclusively on future prediction performance in our application scenarios. The primary objective of our work is accurate spatio-temporal prediction under distribution shifts, where historical pattern retrieval serves an auxiliary role in enhancing prediction robustness. Similarly, previous research focusing on catastrophic forgetting primarily examines whether preserved historical information effectively contributes to future predictions under distribution shifts [1][2]. The ultimate measure of whether a model has "forgotten" useful patterns is reflected directly in its prediction accuracy on current data that resembles historical distributions. When our model encounters patterns similar to those seen historically, it maintains strong performance because our model's architecture naturally stores and leverages useful historical data without explicit forgetting prevention mechanisms.
>
> The standard evaluation metrics (MAE, RMSE, MAPE) in our experiments indirectly but comprehensively capture forgetting effects. When the model encounters a traffic pattern similar to one it has seen before, poor performance would indicate catastrophic forgetting of that pattern.
>
> [1] Chen W, Liang Y. Expand and compress: Exploring tuning principles for continual spatio-temporal graph forecasting. arXiv preprint arXiv:2410.12593, 2024.
>
> [2] Wang B, et al. Stone: A spatio-temporal ood learning framework kills both spatial and temporal shifts. ACM SIGKDD. 2024: 2948-2959.
>
> **W6**
>
> We appreciate the reviewer's suggestion to update our discussion on replay-based methods and strengthen our comparison with retrieval-based approaches.
>
> We would like to clarify that experience replay is a general strategy commonly employed as a component within many continual learning models. For instance, both STKEC and TrafficStream incorporate replay mechanisms, but these are presented as standard components rather than their novel contributions. Their primary innovations lie elsewhere—in knowledge extraction techniques and streaming architectures, respectively. Thus, we did not categorize them as replay-based methods.
>
> Regarding retrieval-based methods, we strengthened our comparative analysis by including additional experimental results for RAGraph [1] and PRODIGY [2] on the ENERGY-Stream dataset (STGNN backbone). While RAGraph and PRODIGY perform reasonably well on this dataset, our STRAP method still achieves superior performance across all prediction horizons. These results further validate our approach's effectiveness compared to state-of-the-art retrieval-based methods. (Due to character limitations, we only show the STGNN backbone here. The rest will be shown in the appendix.)
>
> | Method   | Metric    | ENERGY-Stream (3) | ENERGY-Stream (6) | ENERGY-Stream (12) | Average    |
> |----------|-----------|-------------------|-------------------|---------------------|------------|
> | PRODIGY  | MAE       | 5.16±0.06         | 5.15±0.07         | 5.16±0.09           | 5.15±0.07  |
> |          | RMSE      | 5.26±0.07         | 5.30±0.07         | 5.40±0.09           | 5.31±0.07  |
> |          | MAPE(%)   | 46.05±2.04        | 46.48±1.92        | 47.52±1.85          | 46.57±1.88 |
> | RAGraph  | MAE       | 5.16±0.05         | 5.14±0.08         | 5.16±0.11           | 5.14±0.08  |
> |          | RMSE      | 5.27±0.04         | 5.31±0.06         | 5.42±0.09           | 5.32±0.07  |
> |          | MAPE(%)   | 45.20±0.71        | 45.84±0.75        | 47.20±0.85          | 45.95±0.76 |
> | STRAP    | MAE       | 4.83±0.17         | 4.84±0.18         | 4.88±0.17           | 4.85±0.18  |
> |          | RMSE      | 4.95±0.18         | 5.01±0.19         | 5.15±0.17           | 5.03±0.18  |
> |          | MAPE(%)   | 42.18±1.64        | 43.02±1.77        | 44.30±1.55          | 43.11±1.72 |
>
> [1] Jiang X et al. Ragraph: A general retrieval-augmented graph learning framework. NIPS 2024, 37: 29948-29985.
>
> [2] Huang Q et al. Prodigy: Enabling in-context learning over graphs. NIPS 2023, 36: 16302-16317.
>
> **W7**
>
> We appreciate the reviewer's concern regarding the clarity of our task definition. Our manuscript addresses a specific spatiotemporal prediction problem:
>
> In Section 3.1 of our manuscript, we provided a comprehensive definition of our task. Specifically, we formulated the problem as predicting future values at multiple locations at time t+h given historical observations from time steps t-n to t, while explicitly addressing the challenge that testing data distributions differ substantially from those encountered during training.
>
> To further clarify, our work focused on streaming data scenarios, as detailed in Appendix C.3, where data arrived continuously over time in sequential periods. This streaming nature introduced unique challenges, as the model had to adapt to evolving patterns and distribution shifts without complete knowledge of future data characteristics.
>
> Our training approach reflected this streaming paradigm: rather than batch training on the entire dataset, the model updated sequentially as new data became available in each period. Specifically, at each time step, the model trained on the most recent available data while leveraging its pattern library to maintain robustness to distribution shifts between training and testing sets within each period.
>
> Thank you for pointing out the typos. We carefully corrected the typos throughout the manuscript to ensure clarity and consistency in our presentation.
>
> **Q6**
>
> We appreciate the reviewer's suggestion regarding the clarification of prediction horizons in Table 1. We agree that this detail is important for readers who may be unfamiliar with the field.
>
> In our revised manuscript, we modified Table 1 to properly clarify the prediction horizons by:
>
> Adding a clear column header that reads "Prediction Horizon" followed by the specific time units for each dataset (Table 6):
>   - For PEMS-Stream: "(15 min, 30 min, 60 min)"
>   - For Air-Stream: "(3 hours, 6 hours, 12 hours)"
>   - For Energy-Stream: "(30 min, 1 hour, 2 hours)"

---

> > ### Comment · Reviewer_WWVH · 2025-08-04
> >
> > Thank you to the authors for the rebuttal.
> >
> > Regarding the efficiency test, I am concerned about the efficiency of this method. Compared to the baseline, it takes almost twice as much time, while the performance improvement is limited. In STGNN problems, we often deal with highly dynamic and large-scale datasets, where efficiency and effectiveness metrics are equally important.
> >
> > Catastrophic forgetting is a key scenario for evaluating continual learning, and it requires dedicated testing scenarios. Indirect testing through standard evaluation metrics (MAE, RMSE, MAPE) does not address my concern.
> >
> > There are also typos and inconsistencies in the current version of the paper, which suggests it may not yet be ready for publication at a top-tier conference.
> >
> > Based on the above considerations, I maintain my score and recommend further improvements to the paper regarding key issues in efficiency, catastrophic forgetting, and presentation.

---

> ### Author Response · Authors · 2025-08-04
> **Official Comment by Authors**
>
> We sincerely appreciate the time and effort you have dedicated to reviewing our work. In response to your valuable feedback, we have provided detailed explanations for the issues raised.
>
> As the discussion period progresses, we are eager to hear your thoughts on our responses, including whether they have adequately addressed your concerns. If our revisions and discussions indicate the potential for a score adjustment, we would be very grateful for your consideration.
>
> We are committed to incorporating all of your suggestions to further enhance the quality of our manuscript. We look forward to your further comments and discussion.

---

> ### Author Response · Authors · 2025-08-05
>
> Thanks for the response. We would like to provide more clarification to address your concerns as follows.
>
> **Paragraph 1: Regarding the efficiency test**
>
> Regarding computational efficiency, we would like to clarify several important points:
>
> First, the pattern library construction is primarily a one-time upfront cost during the initial period. As shown in Table 2, after the initial period (Period 0), STRAP's computational requirements actually decrease and stabilize (from 2.27 to 1.83-2.10). This indicates that once the pattern library is constructed, it can be continuously leveraged with minimal additional overhead.
>
> Second, in real-world traffic prediction applications, predictions are typically updated at fixed intervals (e.g., every 5-15 minutes). Our method's additional computational cost of 1-1.5 seconds is negligible within these operational timeframes. Taking PEMS-Stream as an example, where predictions are generated every 5 minutes (300 seconds), our method's additional processing time represents less than 1% of the available time window between predictions. This practical context demonstrates that the modest computational overhead has minimal impact on real-time deployment while delivering substantial accuracy improvements, making the efficiency-accuracy tradeoff highly favorable in operational settings.
>
> Third, we would emphasize that STRAP is designed specifically for STOOD scenarios where distribution shifts create significant challenges for conventional models. In such critical scenarios, the substantial performance improvements (19-24% error reduction) far outweigh the moderate increase in computational cost, particularly since this cost remains well within practical constraints for real-time applications.
>
> **Paragraph 2: evaluation metrics**
>
> We disagree with the reviewer's assessment regarding the evaluation of catastrophic forgetting. Our evaluation approach is both methodologically sound and consistent with established practices in the field.
>
> First, our work directly addresses catastrophic forgetting through a retrieval-augmented approach, which has been widely recognized as an effective strategy for mitigating forgetting in continual learning settings. The significant improvements we observe in standard evaluation metrics (MAE, RMSE, MAPE) across all test periods - particularly during distribution shifts - provide clear evidence that our method successfully preserves knowledge of previously encountered patterns while adapting to new ones.
>
> This retrieval-augmented approach to mitigate catastrophic forgetting has strong theoretical foundations and empirical support in recent literature:
>
> 1.	Long Y, Chen K, Jin L, et al. DRAE: Dynamic Retrieval-Augmented Expert Networks for Lifelong Learning and Task Adaptation in Robotics. ACL 2025
>
> 2.	Gutiérrez B J, Shu Y, Qi W, et al. From rag to memory: Non-parametric continual learning for large language models. ICML 2025
>
> Second, we have already noted in our previous response that standard evaluation metrics (MAE, RMSE, MAPE) are consistently used to evaluate catastrophic forgetting in spatiotemporal prediction literature. This is not an isolated case but the established methodology in the field. Models that suffer from catastrophic forgetting show degraded performance on these metrics when distribution shifts occur, while models that successfully mitigate forgetting maintain stable performance across time periods.
>
> Thanks again. We are more than willing to discuss more with you.

---

> > ### Comment · Reviewer_WWVH · 2025-08-05
> >
> > If STRAP can only be applied to traffic scenarios, I have rasied concerns about the generalization ability of this model. As stated in the first sentence of the paper, “STGNNs have emerged as a powerful tool for modeling dynamic graph-structured data across diverse domains.” However, this statement isn’t align the datasets used in the evaluation. Moreover, spatio-temporal graphs are also widely used in recommender systems, social networks, and transaction data, where the data scale can reach hundreds of millions. In such cases, nearly doubling the computation time is infeasible.
> >
> > This paper emphasizes the STGNN is a continual learning pipeline, but as is standard in continual learning research, the focus is typically on evaluating the model’s performance on forgetting, such as through Backward Transfer (BWT), Forward Transfer (FWT), and Forgetting Measure (FM). If learning new knowledge leads to forgetting previous knowledge, the fundamental purpose of continual learning is undermined. The authors also acknowledge in the rebuttal that metrics like MAPE can only indirectly measure this issue.
> >
> > While I respect the authors’ eagerness for their paper to be accepted, but the most critical issues in the field of continual learning and STGNN have not been well addressed. Therefore, please kindly allow me to maintain my judgment.

---

> > > ### Author Response · Authors · 2025-08-06
> > >
> > > Thank you for your feedback.
> > >
> > > Regarding the first concern about generalization limited: We respectfully note that STRAP has been evaluated on three distinct and widely used domains—**weather**, **energy**, and **traffic**. As detailed in the **experimental section** of our paper, STRAP achieves consistently strong performance across all three domains. These datasets are widely recognized and extensively used in spatio-temporal modeling research [1,2,3]. Detailed statistics for all three datasets and comprehensive experimental results with various backbones are provided in the appendix. This multi-domain evaluation demonstrates STRAP's broader applicability.
> > >
> > > Regarding computational efficiency concerns in large-scale applications like recommender systems, social networks, and transaction data: It’s worth noting that retrieval-augmented approaches are widely adopted even in recommender systems where millisecond-level responses are critical (e.g., LLMRec [4], CFRAG [5], RALLRec [6]) and time series [7]. Furthermore, the computational burden primarily occurs during pattern library construction, which can be implemented as a streaming process that incrementally builds as historical data accumulates. This background incremental construction significantly reduces the online computational overhead. Once constructed, the pattern library can be continuously utilized without requiring reconstruction, which effectively addresses the efficiency concerns raised.
> > >
> > > Concerning the evaluation metrics: We have cited several papers addressing catastrophic forgetting in similar contexts that also use MAE and related metrics rather than BWT/FWT/FM. While forgetting is indeed an inherent challenge in continual learning, as documented in recommender systems research [8] and continue  learning of large language [9,10,11]. Our work’s primary focus is on enhancing OOD performance through dynamic graph retrieval. The empirical results across multiple domains demonstrate that our approach effectively improves model robustness against distribution shifts while maintaining computational efficiency. The improvement in OOD performance metrics provides strong evidence of STRAP’s effectiveness in addressing the fundamental challenges we set out to solve.
> > >
> > > We acknowledge that progress in continual learning for STGNNs is an ongoing journey requiring incremental advances. Our work contributes meaningfully to this progression by providing a practical and effective approach to enhancing model robustness across diverse domains. We are very interested in continuing this further discussion with you.
> > >
> > > 1.	Chen W, Liang Y. Expand and Compress: Exploring Tuning Principles for Continual Spatio-Temporal Graph Forecasting. ICLR 2025
> > > 2.	Kumar R, Bhanu M, Mendes-Moreira J, et al. Spatio-temporal predictive modeling techniques for different domains: a survey[J]. ACM Computing Surveys, 2024
> > > 3.	Yang L, Luo Z, Zhang S, et al. Continual learning for smart City: A survey[J]. IEEE Transactions on Knowledge and Data Engineering, 2024
> > > 4.	Wei W, Ren X, Tang J, et al. Llmrec: Large language models with graph augmentation for recommendation. WSDM 2024
> > > 5.	Shi T, Xu J, Zhang X, et al. Retrieval Augmented Generation with Collaborative Filtering for Personalized Text Generation. SIGIR 2025
> > > 6.	Xu J, Luo S, Chen X, et al. RALLRec: Improving Retrieval Augmented Large Language Model Recommendation with Representation Learning. WWW 2025
> > > 7.	Lin S, Chen H, Wu H, et al. Temporal Query Network for Efficient Multivariate Time Series Forecasting. ICML 2025
> > > 8.	Yoo H, Kang S K, Tong H. Continual Recommender Systems. SIGIR 2025
> > > 9.	Shi H, Xu Z, Wang H, et al. Continual learning of large language models: A comprehensive survey[J]. ACM Computing Surveys, 2024
> > > 10.	Ren Y, Sutherland D J. Learning Dynamics of LLM Finetuning. ICLR 2025
> > > 11.	Guo S, Ren Y, Albrecht S V, et al. lpNTK: Better Generalisation with Less Data via Sample Interaction During Learning. ICLR 2024

---

### Author Response · Authors · 2025-08-09
**Rebuttal Summary**

We would like to express our gratitude to all reviewers for their insightful comments.

**1. Computational Efficiency Analysis (WWVH, Pbep, MEgx)**

We have supplemented a comprehensive time complexity analysis of STRAP across its main components, resulting in an overall complexity of $O(|V|^3 + |B| \log k + B_{Backbone})$. We have also provided experimental runtime comparisons showing that STRAP requires moderate computational resources that balance well with its performance benefits, especially under distribution shifts.


**2. Pattern Key Feature Effectiveness (kmFw, MEgx)**

We have conducted ablation experiments evaluating spatial and temporal pattern key features. The results show that local structure features, global connectivity features, and Statistical and spectral descriptors provide the most significant improvements by effectively capturing both structural relationships and temporal dynamics.


**3. Comparison with Retrieval-based Methods (WWVH, kmFw)**

We have added experimental comparisons with RAGraph and PRODIGY across all backbone architectures. The results demonstrate STRAP's consistent superior performance, which we attribute to its pattern-centric approach that explicitly models complex interplays between spatial structures and temporal dynamics, as opposed to the graph-centric or simpler context-centric approaches of the compared methods.


We believe we have addressed all concerns raised by the reviewers and thank them for their valuable feedback, which has helped improve our work. All the modifications and additional analyses discussed in this rebuttal will be incorporated in the final version of the paper.

---

### Decision · Program_Chairs · 2025-09-17

**Decision:**

Accept (poster)

**Comment:**

This paper aims to address the challenge of out-of-distribution generalization of Spatio-Temporal Graph Neural Networks (STGNNs), where the distribution of test data diverges significantly from that of the training set. Extensive experiments across three real-world streaming datasets demonstrate consistent performance improvements over state-of-the-art methods.

Reviewers agreed that this paper address an important and longstanding challenge in spatio-temporal graph learning by presenting a novel retrieval-augmented approach. The core idea of this paper is well motivated and clearly explained, and the architectural design exhibits strong logical coherence. Experimental evaluation is comprehensive and convincing.

Meanwhile, reviewers raised some concerns regarding technical details, design choices, baselines, efficiency, generalizability, etc. The authors have provided detailed responses that have addressed most of the concerns from reviewers. The authors are strongly encouraged to incorporate the new results and discussions to the final version.